# Rapid shifting of a deep magmatic source at Fagradalsfjall volcano, Iceland

Sæmundur A. Halldórsson[1 ✉], Edward W. Marshall[1], Alberto Caracciolo[1], Simon Matthews[1], Enikő Bali[1], Maja B. Rasmussen[1,11], Eemu Ranta[1], Jóhann Gunnarsson Robin[1], Guðmundur H. Guðfinnsson[1], Olgeir Sigmarsson[1,2], John Maclennan[3], Matthew G. Jackson[4], Martin J. Whitehouse[5], Heejin Jeon[5], Quinten H. A. van der Meer[1], Geoffrey K. Mibei[1], Maarit H. Kalliokoski[1], Maria M. Repczynska[1], Rebekka Hlín Rúnarsdóttir[1], Gylfi Sigurðsson[1], Melissa Anne Pfeffer[6], Samuel W. Scott[1], Ríkey Kjartansdóttir[1], Barbara I. Kleine[1], Clive Oppenheimer[7], Alessandro Aiuppa[8], Evgenia Ilyinskaya[9], Marcello Bitetto[8], Gaetano Giudice[10] & Andri Stefánsson[1]

Recent Icelandic rifting events have illuminated the roles of centralized crustal magma reservoirs and lateral magma transport[1–4], important characteristics of mid-ocean ridge magmatism[1,5]. A consequence of such shallow crustal processing of magmas[4,5] is the overprinting of signatures that trace the origin, evolution and transport of melts in the uppermost mantle and lowermost crust[6,7]. Here we present unique insights into processes occurring in this zone from integrated petrologic and geochemical studies of the 2021 Fagradalsfjall eruption on the Reykjanes Peninsula in Iceland. Geochemical analyses of basalts erupted during the first 50 days of the eruption, combined with associated gas emissions, reveal direct sourcing from a near-Moho magma storage zone. Geochemical proxies, which signify different mantle compositions and melting conditions, changed at a rate unparalleled for individual basaltic eruptions globally. Initially, the erupted lava was dominated by melts sourced from the shallowest mantle but over the following three weeks became increasingly dominated by magmas generated at a greater depth. This exceptionally rapid trend in erupted compositions provides an unprecedented temporal record of magma mixing that filters the mantle signal, consistent with processing in near-Moho melt lenses containing $10^7$–$10^8$ m$^3$ of basaltic magma. Exposing previously inaccessible parts of this key magma processing zone to near-real-time investigations provides new insights into the timescales and operational mode of basaltic magma systems.

Geological, geochemical and geophysical observations at mid-ocean ridges (MOR) and adjacent transform faults are fundamental to understanding oceanic crust formation[1,8]. Generally, oceanic crust formation is associated with active spreading centres along MOR, where eruptions of mid-ocean ridge basalts are fed from crustal reservoirs[8,9]. However, more rarely, MOR eruptions may be supplied from sub-Moho levels (more than 7 km), typically at slow-spreading ridges and transform faults[9–11]. Because of the inaccessibility of deep ocean environments, near-real-time observations and investigations of such events are limited.

Although the Icelandic crust is thicker than typical oceanic crust[1], the subaerial exposure of the MOR permits continuous, real-time sampling of eruptions yielding critical insights into magma processes and timescales representative of their submarine counterparts. Two Icelandic rifting events—the 1975–84 Krafla[1] and the 2014–2015 Bárðarbunga[2,3] eruptions—have been documented by geophysical monitoring and contemporaneous lava sampling, providing near-real-time insights into the magmatic plumbing system. Both eruptions revealed the importance of mid to shallow crustal storage—that is, the topmost 10 km of the crust—of basalts in centralized magma reservoirs from which melt was transported laterally before eruption[4]. However, these shallow magmatic processes obscured the record of deeper interactions. The Reykjanes Peninsula (RP) is a subaerial trans-tensional oblique rift, or a so-called leaky transform fault (Fig. 1a), characterized by episodic rifting and associated volcanism at approximately millennial intervals[12–14]. Until 2021, the last eruptive episode on the RP occurred between circa AD 700 and circa AD 1240, when four of five RP intratransform spreading centres (ITSC) erupted[12–14]. The last eruptions on the fifth of these ITSCs in Fagradalsfjall (Fig. 1) took place more than 6,000 yr BP[13].

The 2021 eruption in the Fagradalsfjall complex (Fig. 1b) was preceded by tectonic unrest from January 2020, with intense earthquake activity and inflation centred on the Svartsengi ITSC. Seismicity

[1]Nordic Volcanological Center, Institute of Earth Sciences, University of Iceland, Reykjavík, Iceland. [2]Laboratoire Magmas et Volcans, Université Clermont Auvergne, Aubière, France. [3]Department of Earth Sciences, University of Cambridge, Cambridge, UK. [4]Department of Earth Science, University of California Santa Barbara, Santa Barbara, CA, USA. [5]Department of Geosciences, Swedish Museum of Natural History, Stockholm, Sweden. [6]Icelandic Meteorological Office, Reykjavík, Iceland. [7]Department of Geography, University of Cambridge, Cambridge, UK. [8]Dipartimento di Scienze della Terra e del Mare, Università di Palermo, Palermo, Italy. [9]COMET, School of Earth and Environment, University of Leeds, Leeds, UK. [10]Istituto Nazionale di Geofisica e Vulcanologia, Catania, Italy. [11]Present address: Department of Geosciences and Natural Resource Management, University of Copenhagen; 1350, Copenhagen, Denmark. ✉e-mail: saemiah@hi.is

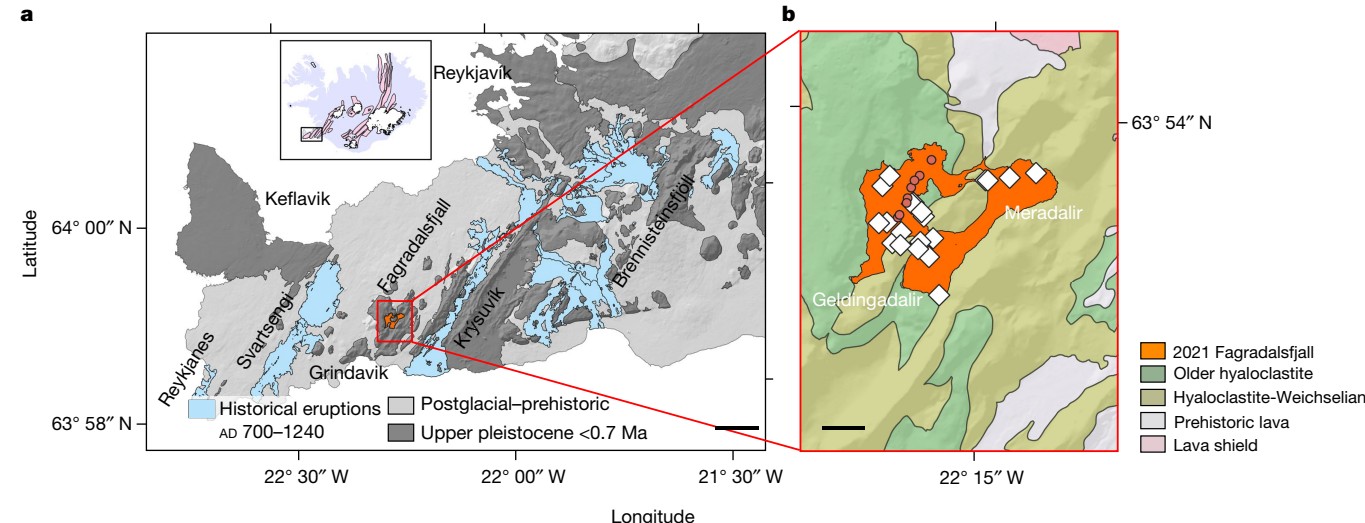

**Fig. 1 | Geological setting. a**, Geological setting of the RP and the ITSCs of Reykjanes, Svartsengi, Fagradalsfjall, Krýsuvík and Brennisteinsfjöll extending from west to east are shown in bold[13]. The inset map of Iceland shows the location of the RP. Scale bar, 5 km **b**, The Fagradalsfjall ITSC and the eruption sites. The extent of the lava field corresponds to 10 May 2021[16]. The eruptive vents are shown with red circles. Sampling localities are also shown with white diamonds. Scale bar, 1 km.

remained episodic during 2020, with the latest unrest cycle initiated by a magnitude 5.7 earthquake on 24 February 2021. After three weeks of intense seismicity, surface deformation indicated rifting accompanying a dyke injection along normal faults in the vicinity of Fagradalsfjall[15]. The intrusion reached the surface on 19 March 2021. The first several weeks of the eruption were characterized by a low to modest magma effusion rate of 1–8 $m^3 s^{-1}$ from multiple vents, associated with formation of spatter cones and low-viscosity lava flows. The eruption style changed after April 27th with an increased discharge rate (9–13 $m^3 s^{-1}$) from a single vent and high (more than 450 m) lava fountaining[16].

Here we focus on the initial phase of the eruption (21 March to 6 May 2021) and combine geochemical analyses of sequentially erupted lava and tephra samples with measurements of vent gas emissions. Whole-rock and glass major and trace element contents, as well as radiogenic isotope ratios, were measured along with the mineral phases and their melt inclusions (MIs) (Methods). The Fagradalsfjall lavas are olivine tholeiite basalts with petrographic features (Extended Data Fig. 1) and major element compositions similar to RP lavas erupted historically (that is, since settlement, circa AD 870)[14,17,18] (Fig. 2a). Whole-rock MgO and $TiO_2$ contents range from 8.8 to 10.0 wt% and from 0.95 to 1.12 wt%, respectively (Extended Data Fig. 2a), for which the high-MgO content suggests the magmas were less processed in shallow magmatic plumbing systems, and here these are referred to as 'primitive'. Glasses from quenched lava and tephra, which have MgO contents between 6.7 and 9.0 wt% and $TiO_2$ contents between 0.83 and 1.54 wt% (Extended Data Fig. 2a), define a more evolved (that is, less primitive) and chemically variable group than the whole-rock samples, largely reflecting crystallization within the conduit and lava flow. The primitive character of the Fagradalsfjall lava is further indicated by its primitive crystal cargo, which is typical of magmas that have experienced limited low-pressure magmatic evolution. For example, macrocrysts have highly primitive cores: Cr-rich spinel with Cr# up to 50.2, $Fo_{90}$ olivine, $An_{91}$ plagioclase and green clinopyroxene with Mg# up to 88.8 and $Cr_2O_3$ up to 1.48 wt% (Extended Data Fig. 3; Supplementary Information).

Sampling of primitive magmas is not uncommon on the RP, and lavas with MgO contents above 8 wt% and primitive crystal cargoes are represented among historical lavas. In contrast to the RP lavas, the Fagradalsfjall lava displays an exceptionally wide range in ratios of incompatible minor elements $K_2O$ and $TiO_2$ (0.124–0.263; Fig. 2a). $K_2O/TiO_2$ and other incompatible trace element ratios (ITERs) are insensitive to variations in modal crystal proportions and thus fractional crystallization. Although interactions between the magma and some magmatic phases (for example, titano-magnetite and amphibole) can fractionate $K_2O/TiO_2$, such phases are unlikely to be present in substantial proportions in the magma storage region. Moreover, in contrast with single-eruptive basaltic units (MgO > 6.5 wt%) from different parts of the Icelandic rift system, for which large datasets are available, the Fagradalsfjall lava has an uncommonly wide range in $K_2O/TiO_2$ values (Fig. 2b). Notably, it has been suggested that high-MgO units best preserve signatures associated with the Icelandic mantle (for example, Borgarhraun, North Iceland)[19].

The large range in $K_2O/TiO_2$ became apparent in both whole-rock and glass samples as the eruption progressed. In only the first three weeks of the eruption, the Fagradalsfjall lava underwent a remarkable compositional shift in $K_2O/TiO_2$ and La/Yb—which are geochemical proxies that signify different mantle compositions and/or melting conditions—with both increasing by a factor of around 2 (Fig. 3). The Fagradalsfjall lava also records a simultaneous shift towards more radiogenic Sr and Pb, and less radiogenic Nd isotope ratios, confirming that the deeply derived, lower-degree melts from later in the eruption sample a higher proportion of an enriched mantle source with higher incompatible trace element concentrations, and radiogenic isotope signatures indicative of long-term incompatible trace element enrichment (Extended Data Fig. 4). Clear correlations ($R^2$ > 0.97) between $K_2O/TiO_2$, La/Yb and Pb isotopes confirm that the large range in Fagradalsfjall $K_2O/TiO_2$ reflects mantle-derived variability. Strikingly, over those first three weeks, the Fagradalsfjall lava composition encompassed and exceeded the entire spectrum of mantle source indicators (for example, ITERs and radiogenic isotopes) of lavas erupted from the around 20 volcanic fissures active across the RP during the last approximately 540-year-long (around AD 700 to 1240) rifting episode (Extended Data Fig. 4)[14,17,18].

The remarkable compositional diversity of the Fagradalsfjall eruption is further reflected by primitive MI compositions. Notably, MIs with high-MgO contents (more than 10 wt%, corrected for post-entrapment processes (PEPs); Methods) also display a wide range in $K_2O/TiO_2$ and La/Yb, confirming the presence of diverse mantle melts within a magma reservoir during host mineral crystallization[7,20] (Extended Data Fig. 2c,d). Both $K_2O/TiO_2$ and La/Yb are insensitive to PEP corrections and, therefore, the large variability present—even within a single crystal (Extended Data Fig. 2b)—must reflect primary variability in the magma

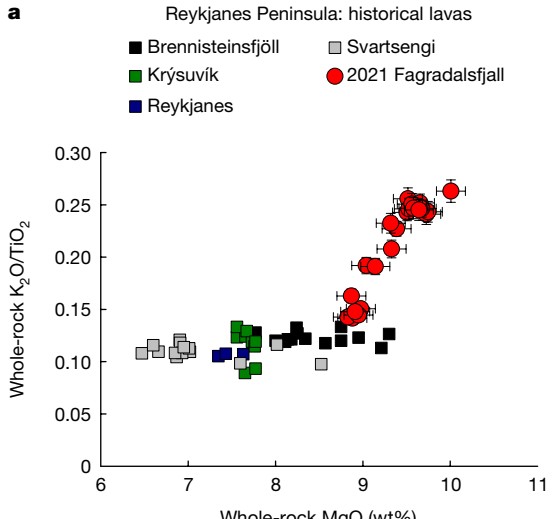

**a** Reykjanes Peninsula: historical lavas
- ■ Brennisteinsfjöll  ▪ Svartsengi
- ■ Krýsuvík   ● 2021 Fagradalsfjall
- ■ Reykjanes

**b** Iceland: single eruptive units
- ▲ Borgarhraun  ■ Thjofahraun  ● 2021 Fagradalsfjall
- ▽ Laki   ■ Hallmundarhraun
- ■ Lambahraun  ▽ 2014–15 Holuhraun

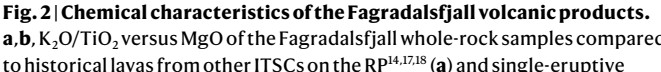

**Fig. 2 | Chemical characteristics of the Fagradalsfjall volcanic products.**
**a**,**b**, $K_2O/TiO_2$ versus MgO of the Fagradalsfjall whole-rock samples compared to historical lavas from other ITSCs on the RP[14,17,18] (**a**) and single-eruptive Icelandic basaltic lavas from different parts of the rift system for which large datasets are available (ref. [3] and references therein) (**b**). Error bars are included on both panels and include external $2\sigma$ error.

reservoir(s) that fed the eruption (including integrated crystal mush horizons[7]). Most of the MIs are geochemically depleted, extending to compositions thought to represent the depleted parental melt end member of the RP[21]. However, a small subset of enriched olivine-hosted MIs, sampled in the early comparatively depleted carrier melts and whole rocks, have elevated $K_2O/TiO_2$ up to 0.39. The entrapment of high $K_2O/TiO_2$ melts indicates that the enriched mantle melt—which came to dominate lava compositions as the eruption progressed—was already present in the magmatic system, but probably partially homogenized with depleted melts before the onset of the eruption. Even though wide compositional diversity is typical of primitive MIs associated with basaltic eruptions globally[8], the Fagradalsfjall eruption is a basaltic eruption in which such extreme mantle-derived compositional diversity is observed in whole-rock and glass compositions, in addition to its MIs, in near-real-time.

Thermobarometry indicates that the eruption is sourced from a near-Moho[22] magma reservoir at a depth of more than 15 km (more than 0.4 GPa) (Fig. 4a and Extended Data Figs. 5 and 6; Methods). Tephra from the fountaining stage of the eruption (samples from 28 April 2021 and onwards), in addition to primitive (MgO > 9 wt%) MIs and cores of primitive ($Cr_2O_3$ > 1 wt%) clinopyroxene macrocrysts (Supplementary Information) throughout the studied period of the eruption, yield pressure estimates from 0.36 to 0.80 GPa, with the most probable pressure range being 0.55–0.65 GPa (equivalent to a depth of around 20 km). A similarly deep magma provenance is indicated by a combined analysis of volatiles in MIs and in surface vent gas emissions, which demonstrate that magma ascended from a depth of 19 ± 4 km depth (Fig. 4a and Extended Data Figs. 7–10; Methods). By contrast, the earlier erupted tephra and groundmass glass, and evolved clinopyroxene macrocryst rims and cores, equilibrated at shallower, crustal depths (0.05–0.25 GPa; a depth of less than 8 km) (Extended Data Figs. 5 and 6). Thus, although minor compositional modification of the carrier liquids and growth of crystal rims occurred in transit during decompression, mixing and cooling, high-pressure phase relationships of erupted products were preserved and suggest deep, near-Moho magma storage.

The compositional diversity of the Fagradalsfjall lava is thus consistent with a derivation from aggregation of melts (that is, magma reservoirs) located within the mantle or close to the crust–mantle boundary, progressively acquiring melt from greater depths (Fig. 4). The presence of some macrocrysts too primitive to have crystallized from the Fagradalsfjall carrier liquid directly is consistent with an

accumulated cognate load of crystals derived from mushes at the margins of a melt lens. Geochemical diversity in melts under Iceland is thought to be generated by progressive polybaric near-fractional melting of the lithologically heterogeneous mantle[7,14,18]. Moreover, the mean composition of mantle melts being supplied to the base of the crust varies as a result of spatial and temporal fluctuations in mantle decompression rates and melt supply through porous flow in channels (for example, refs. [7,18]). At Fagradalsfjall, the early erupted material has a composition weighted towards that expected for melts generated in the shallow parts of the melting region (Extended Data Fig. 11). The mean composition of diverse MI suites in Iceland has been shown to track the long-term mean composition of mantle melts supplied to the base of the crust[7,20]. The Fagradalsfjall MIs record a mean composition more depleted than that seen at the start of the eruption (Fig. 3a,b), indicating that addition and mixing of the enriched end member had already begun before the eruption onset. It is, therefore, possible that this mixing event may have triggered magma ascent into the shallower crust and eventual eruption.

As the Fagradalsfjall eruption progressed, the melts became increasingly enriched, and evolved to higher La/Yb ratios, consistent with a gradual incorporation of deeper-generated mantle melts[18] into the magma reservoir sourcing the Fagradalsfjall eruption (Extended Data Fig. 11). This resulted in the gradual increase in $K_2O/TiO_2$, La/Yb and $^{206}Pb/^{204}Pb$ (Figs. 3 and 4b). These incoming, deeply formed, enriched melts would dominate later in the eruption (mid-April to early May). They may either have been directly supplied from the high-flux centre of a mantle melt supply channel, weighted towards deep melts[23], or by drainage of a second melt lens containing geochemically enriched liquids in the proximity of the first-tapped depleted lens. Drainage of a proximal enriched magma body would be consistent with the high pressure estimates recorded for tephra collected from the fountaining stage of the eruption (0.6 GPa; around 22 km; Fig. 4a and Extended Data Fig. 6).

Similar mantle-derived variability has been observed in some petrologically well-characterized single-eruptive oceanic basalt units mapped and sampled for within-flow variations, consistent with incomplete mixing and variable differentiation before eruption[8]. However, a comparison of the Fagradalsfjall $K_2O/TiO_2$ variability—which is a robust proxy for global geochemical enrichment[24]—with several of those units also highlights the unique case of the Fagradalsfjall lava (Extended Data Fig. 12). For example, individual MOR basaltic eruptions from

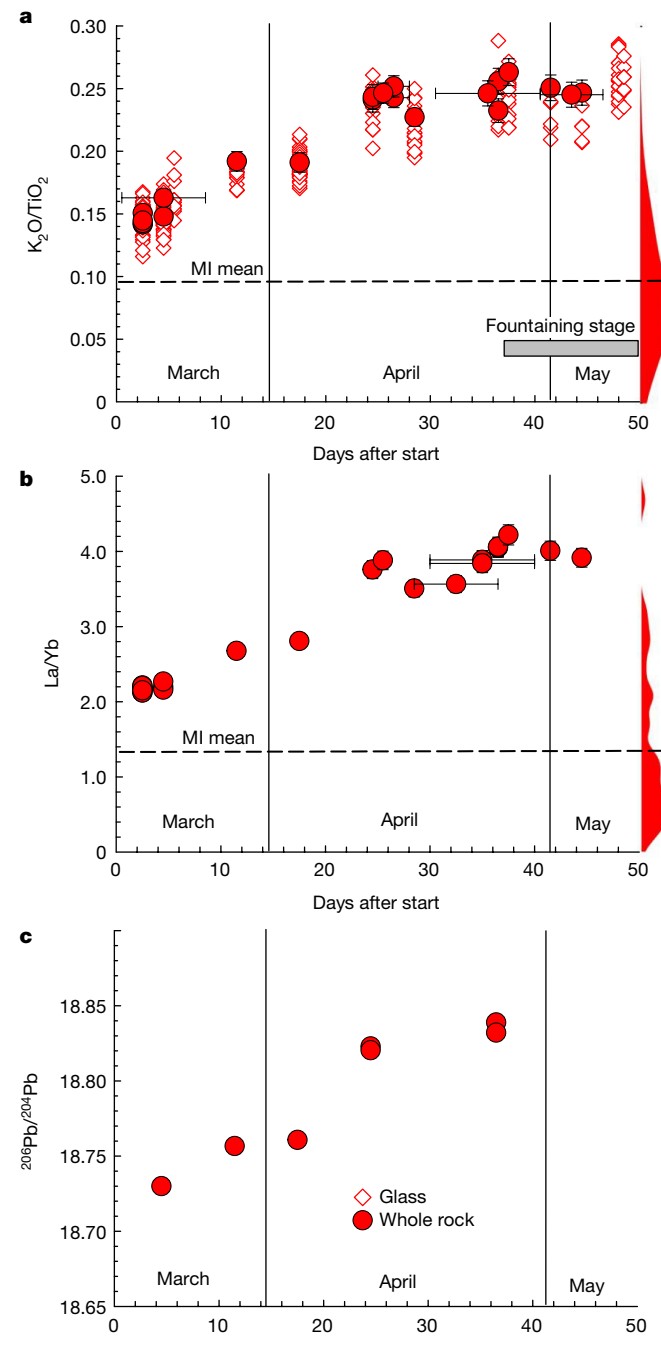

**Fig. 3 | Temporal trends evident over the course of the first 50 days of 2021 Fagradalsfjall eruption. a–c**, K$_2$O/TiO$_2$ (**a**), La/Yb (**b**) and $^{206}$Pb/$^{204}$Pb (**c**) versus days after start of the eruption. The kernel density estimates on the edges of **a** and **b** show the distribution of measured K$_2$O/TiO$_2$ and La/Yb for Fagradalsfjall MIs and the dashed line is the MI mean. The bandwidths are estimated using Scott's rule: 0.14 for La/Yb and 0.036 for K$_2$O/TiO$_2$. Error bars are included and indicate external 2$\sigma$ error for geochemical data and a possible range of eruption days when not known precisely. Error bars for $^{206}$Pb/$^{204}$Pb are generally smaller than the symbol. For glass data error estimation, see Extended Data Fig. 2c.

(1) plume-influenced ridges along the Galápagos spreading centre[25], (2) near 17.5° S on the East Pacific Rise (EPR)[26], (3) newly formed EPR (2005–2006) eruptions[27] and (4) recent eruptions on the Axial seamount[28] are all characterized by nearly uniform K$_2$O/TiO$_2$. Even sites

where single lava flows (for example, the N1 unit at 17.5° S on the EPR) show both considerable radiogenic isotope and incompatible trace element heterogeneity[26] reveal limited K$_2$O/TiO$_2$ variability in comparison with Fagradalsfjall (Extended Data Fig. 13). This is notable in light of the Fagradalsfjall eruption being mainly fed from a single vent, in contrast to the EPR eruptions that can be fed from multiple segmented melt lenses extending over many kilometres, resulting in variable processing of magmatic compositions[9,26].

The frequent sampling of lavas during the eruption allows evaluation of the rate of geochemical change. In comparison to other eruptions for which high-resolution temporal data are available[29–31], the Fagradalsfjall eruption has significantly greater rates of change in terms of mantle source indicators. For example, in comparison with Kīlauea's Puʻu ʻŌʻō 1983–2018 eruption (Extended Data Fig. 12a), the Fagradalsfjall eruption shows larger geochemical source change over a shorter period of time. To show this, we quantify the rate of compositional shift for the Puʻu ʻŌʻō eruption as d$R$/d$t$, where $R$ represents the ratio of two incompatible elements or isotopes and $t$ is time. Using the highest and lowest K$_2$O/TiO$_2$ values (0.19 in January 1983 and 0.17 in January 2015) as an example[29,31] we calculate a d$R$/d$t$ (0.02/32 yr) for Puʻu ʻŌʻō of $6.3 \times 10^{-4}$ yr$^{-1}$. In comparison, the rate of the source signature shifts at Fagradalsfjall (d$R$/d$t$ = 0.1/0.12 yr) is over four orders of magnitude faster, or 0.81 yr$^{-1}$. For Pb isotopes ($^{206}$Pb/$^{204}$Pb), the comparison is similar: $6.3 \times 10^{-4}$ yr$^{-1}$ for Puʻu ʻŌʻō and 1 yr$^{-1}$ for Fagradalsfjall.

The observation that the change in the Fagradalsfjall lava composition is gradual and that the variability in samples over timescales of approximately five days is limited (Fig. 3) indicates effective homogenization of the melt before its supply to the surface. Simple fluid dynamics models of basaltic sills in the Icelandic lower crust indicate that homogenization of passive tracers (for example, ITER and Pb isotopes) occurs within a few hours[7], suggesting that the around 20 day gradual compositional transition seen at Fagradalsfjall (Fig. 3) does not reflect internal rearrangement of the heterogeneity initially present in a sill. Gradually increased supply of enriched melt from a second discrete sill or from a mantle melt supply channel during the eruption could explain the observations. In this case, and equating the melt supply with the observed lava effusion rate, the timescale of geochemical evolution of the lavas is given by the ratio of reservoir volume to effusion rate. Taking the latter as 10 m$^3$ s$^{-1}$ (ref. 16), and for a timescale of 20 days, the initial melt lens volume was of the order of $2 \times 10^7$ m$^3$. This is equivalent to an approximately 6-m-thick disc of radius 1 km, which is consistent with estimates considering changes in effusion rates during the eruption[16] and conceptual models of sill-like melt storage in the Icelandic lower crust[7]. This simple model captures the rapid change in melt compositions, and predicts that the geochemically depleted end-member melt (present at the start of the eruption) was largely drained from a melt lens, which was replenished with enriched melt within 20 days (Fig. 4c).

Primitive lavas of the Fagradalsfjall eruption present a window into the deep roots of a magmatic system previously inaccessible to near-real-time investigation. The eruption was fed directly from a near-Moho reservoir without protracted stalling and equilibration at shallow levels in the crust, which is consistent with the scenario inferred for other primitive prehistoric Icelandic eruptions[7,19,32]. This eruption revealed extreme mantle-derived compositional heterogeneity, with the initially tapped magma replaced with a deeper-derived melt in only a few weeks. Although magma mixing in the deep crust and uppermost mantle has been inferred through MIs and crystal chemistry studies[3,7,8,20], this is a direct observation of magma mixing within the mantle. Critically, studying mantle magma mixing through minerals and their MIs provides only indirect and model-dependent timescale and volumetric information. The Fagradalsfjall eruption shows that eruptible batches of basaltic magma mix on a timescale of weeks at the crust–mantle boundary.

The Fagradalsfjall eruption clearly demonstrates how rapidly a deep magmatic plumbing system can reconfigure its eruptive outputs.

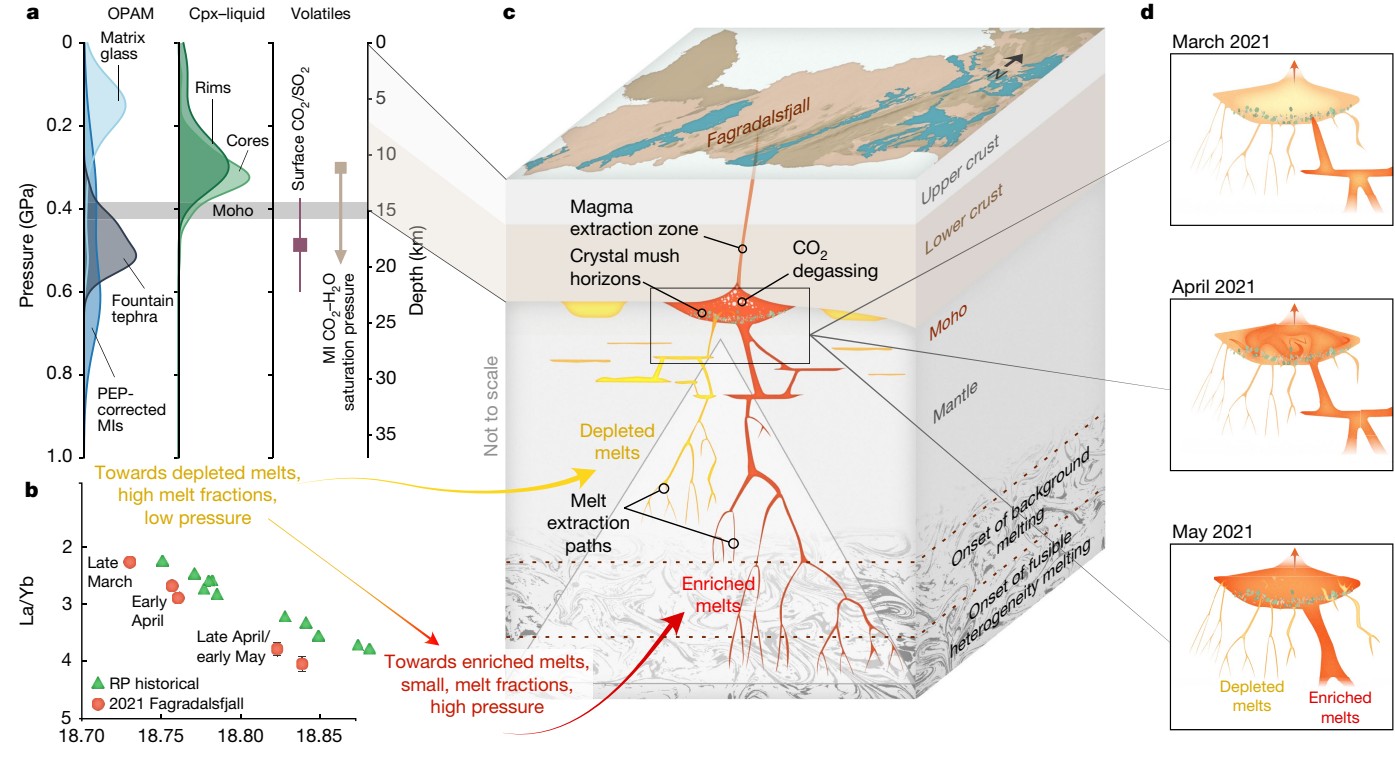

**Fig. 4 | Conceptual model of melt extraction, accumulation, mixing and crustal ascent beneath Fagradalsfjall. a**, Melt storage pressures obtained by olivine-plagioclase-augite-melt (OPAM) barometry using compositions from glass, and MIs, clinopyroxene (cpx)–liquid barometry from crystal cores and rims, and the storage pressures consistent with the gas $CO_2/SO_2$ ratio, assuming closed-system degassing. The curves are kernel density estimates produced using a bandwidth based on the number of data points (Scott's rule), which in all cases was greater than the measurement uncertainty. **b**, The lava erupted at the start of the eruption was depleted in composition, consistent with shallow, high-degree melting of a relatively depleted mantle source (yellow). However,

as the eruption progressed, the melts became increasingly enriched, consistent with deeper, lower-degree partial melting of a more enriched mantle source (red). Note the reversed axis for La/Yb. **c**, A conceptual model of melt extraction, accumulation and crustal ascent beneath Fagradalsfjall. Melts are generated in the mantle and ascend to a near-Moho storage zone where crystallization, mixing and degassing occur before eruption. **d**, Evolution of a near-Moho reservoir that explains the erupted lava compositions at Fagradalsfjall. Initially, the magma reservoir contained depleted melt, but over the course of the eruption continuous recharge of enriched melt resulted in a compositional change within the magma reservoir.

It thereby provides observations critical for testing models of transcrustal magmatism, including the timescales of reconfiguration of melt supply systems[6,7] and physical models of magmatic systems. For example, how might models dominated by low-porosity mushes account for the rapid transition in erupted composition at Fagradalsfjall?[33] Our observations provide a rare 'snapshot' of dynamic melt extraction, mixing and aggregation processes that occur near or below the crust–mantle boundary. How widespread such deep magmatic plumbing system reconfigurations are at MOR and other oceanic islands where the crust is thinner remains to be explored.

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

# Methods

## Whole-rock major element, trace element and isotope analysis

**Inductively coupled plasma optical emission spectroscopy.** A set of nine whole-rock (glass, tephra) samples (each sample prepared in triplicate) were analysed for major ($SiO_2$, $Al_2O_3$, FeO, MnO, MgO, CaO, $Na_2O$, $K_2O$, $TiO_2$ and $P_2O_5$) elements using the Inductively Coupled Plasma Optical Emission Spectroscopy (ICP-OES) instrument (ThermoFisher iCAP 7400 Duo) at the Institute of Earth Sciences, University of Iceland. The analytical details have been reported elsewhere[3] and are followed here with some minor exceptions. Portions of fresh lava collected, while hot, at the eruption site were powdered in a tungsten carbide disc mill. Calibration was based on in-house reference samples (A-THO, B-THO and B-ALK), calibrated through repeated analyses of United States Geological Survey (USGS) reference material (BHVO-1, BIR-1, W-2 and BCR-1). On the basis of repeated measurements of the reference material, precision for major and minor elements (% range) is less than ±0.5% relative for major oxides (except for $P_2O_5$) (Supplementary Table 2). As no volatile and redox titration analyses were performed, the chemical composition is reported on a dry basis and was recalculated to 100%, expressing total iron as FeO.

**Inductively coupled plasma mass spectrometry.** See the Supplementary Information for a full method description. Hand-picked groundmass glass, or agate-ground whole rocks, were digested in a HF–$HNO_3$ mixture in Savillex teflon beakers in class-100 laminar flow hoods in the geochemical clean laboratory at the Institute of Earth Sciences, University of Iceland. An aliquot of each digested sample was transferred to an acid-leached 10 ml vial, spiked with an In-Re internal standard, and diluted with 2% $HNO_3$ to a dilution factor of around 5,000. This solution was analysed for trace elements (see Supplementary Table 3 for elements analysed) on the ThermoFisher iCap RQ Quadrupole Inductively Coupled Plasma Mass Spectrometer at the Institute of Earth Sciences, University of Iceland. The data were calibrated using the USGS standards BIR-1, BHVO-2, BCR-2 and AGV-2 using preferred values from the GeoRem database (http://georem.mpch-mainz.gwdg.de). Accuracy was tested by running the W-2 standard as an unknown, which reproduced all elements within 3.6% of the GeoRem value. Precision and reproducibility were tested by measuring separate duplicate digestions of W-2 as an unknown along with the sample analyses ($n = 7$); the relative standard deviation of W-2 duplicates for all measured elements was less than 5% ($2\sigma$) except for Cu, V and U, which were 5.9%, 6.9% and 5.7%, respectively (Supplementary Table 3). Duplicates of picked groundmass glass have slightly worse precision (10%) than W-2 related to the inherent greater heterogeneities of natural volcanic groundmass glass samples.

**Multi-collector inductively coupled plasma mass spectrometry.** See the Supplementary Information for a full method description. Lead isotopes were analysed at the Institute of Earth Sciences, University of Iceland using a Nu Plasma multi-collector inductively coupled plasma mass spectrometer (the procedure was similar to that in ref. [3]). Glass and whole-rock samples were leached by sonicating in 6N HCl for 60 min and were then dissolved in the same manner as for trace element analysis. Ion chromatography for Pb separation was based on the recommendations from ref. [34]. Instrumental mass fractionation was corrected using a Tl spike. During each run, SRM-981 was run as a bracketing standard and all Pb-isotopic ratios were normalized to the SRM-981 values from ref. [35]. AGV-2, BCR-2 and JB-3 were analysed as secondary standards and the results agree with published values to within reported compositional uncertainty (Supplementary Table 4).

**Thermal ionization mass spectrometry.** Sr and Nd isotopes were analysed at Laboratoire Magmas et Volcans at Université Clermont Auvergne using a Triton (ThermoScientific) thermal ionization mass spectrometer. Tephra and lava samples were crushed in a jaw crusher and powdered in an agate bowl using a planetary mill. Approximately 200 mg of powder was dissolved in concentrated HF and $HNO_3$ acids (ratio 3:1). Fluoride residue was reduced by repeated dissolution and evaporation in 6N HCl and 14N $HNO_3$. Isolation of Sr and Nd followed the protocol from ref. [36], and their isotope compositions were measured by thermal ionization mass spectrometry. Strontium isotope ratios were normalized to $^{86}Sr/^{88}Sr = 0.1194$ and to the recommended value of the NIST 987 standard ($^{87}Sr/^{86}Sr = 0.710245$). The USGS AGV-1 reference material was measured five times as a secondary standard and yielded an average $^{87}Sr/^{86}Sr$ of $0.703980 \pm 5$ ($2\sigma$), which is consistent with the literature value of ref. [37], $0.703989 \pm 17$ ($2\sigma$). The $^{143}Nd/^{144}Nd$ and $^{145}Nd/^{144}Nd$ ratios were corrected for time-dependent mass fractionation by normalization to $^{146}Nd/^{144}Nd = 0.7219$, and to $^{143}Nd/^{144}Nd = 0.512100$ for the JNd-1 standard (Supplementary Table 4). Four analyses of the USGS AGV-1 reference material yielded an average $^{143}Nd/^{144}Nd$ of $0.512772 \pm 10$ ($2\sigma$), which is within the uncertainty of the literature value of ref. [37] of $0.512791 \pm 13$ ($2\sigma$).

## Major and minor element analysis of glasses, MIs and minerals

**Electron probe micro-analysis.** Minerals and glasses in 12 round 1-inch epoxy mounts were analysed by the electron probe micro-analyser at the Institute of Earth Sciences, University of Iceland. The JEOL JXA-8230 SuperProbe electron probe micro-analyser is equipped with a thermionic electron emitter and five wavelength-dispersive spectrometers. All phases were analysed with an accelerating voltage of 15 keV. The concentration of each element was derived from counting X-rays at the peak characteristic for each element in the X-ray spectrum and the background on both sides of the peak. For glass analyses, the probe current, measured at the Faraday cup before each analysis, was 10 nA, and the beam diameter was 10 μm for groundmass glasses and 10 μm or 5 μm beam for MIs, depending on their dimensions. Plagioclase was also analysed with a probe current of 10 nA and pyroxene, olivine and spinel with a current of 20 nA, and the beam was either focused or had a 5 μm diameter. The standards used, their origin and the counting times for the peak and background for each element are listed in Supplementary Tables 5–10. In all cases, the data reduction was performed with the CITZAF correction program, except in the case of oxide analyses, for which the ZAF correction was used (see the details in ref. [3]).

## Major volatiles and trace element analysis in MIs

**Secondary ion mass spectrometry.** A detailed description of the Secondary Ion Mass Spectrometer (SIMS) trace element, $H_2O$ and $CO_2$ methods can be found in the Supplementary Information; an abbreviated version is given here. The trace elements in olivine and plagioclase-hosted MIs were measured using a CAMECA IMS1280 SIMS instrument at the NordSIM facility, Swedish Museum of Natural History, Stockholm, following the procedure described in ref. [38]. A Hyperion H201 RF Plasma source was used to generate a 6 nA primary ion beam of $O^-_2$ ions, rastered over a $10 \times 10$ μm$^2$ area during analysis. Overall trace element uncertainties are estimated from the in-run uncertainty on measured $M^+/^{28}Si^{16}O^+_2$ ratios and from the repeatability on standard measurements. Uncertainties varied as a function of concentration: trace elements with concentrations greater than 5 μg g$^{-1}$ typically have uncertainties of less than 10% ($1\sigma$), elements with concentrations greater than 1 μg g$^{-1}$ typically had uncertainties of less than 20% ($1\sigma$) and elements with concentrations between 1 and 0.17 μg g$^{-1}$ typically had uncertainties of less than 30% ($1\sigma$). Secondary standards GOR128-G and GOR132-G produced average concentrations within 20% of accepted values, except for Nb, which suffers from isobaric interference from $^{40}Ca^{53}Cr$ that results in a detection limit of around 0.1 μg g$^{-1}$ on these standards (Nb data are reported but not considered in this study). The contents of $H_2O$ and $CO_2$ were determined in MIs and matrix glass using the same SIMS instrument used for trace elements. A 2 nA $O^-_2$ primary beam was used, with the same raster size as for trace elements. $^1H^+$, $^{12}C^+$

and $^{28}Si^{2+}$ species were measured using a peak hopping routine. Conversion of $^1H^+/^{28}Si^{2+}$ and $^{12}C^+/^{28}Si^{2+}$ ratios to $H_2O$ and $CO_2$ was achieved using a set of volatile doped basaltic glasses[39] (M5, M15, M43, M60 and N72). Uncertainties were calculated from the signal variability of the sample analysis, the scatter along the calibration curve and the uncertainty of the composition of the calibrating standards. Overall analytical uncertainties for $H_2O$ are less than 4%, and less than 8% for $CO_2$, based on repeatability of the calibration reference materials. Basalt glass A35 was measured as a secondary standard and produced average $H_2O$ and $CO_2$ concentrations within 3% of accepted values (Supplementary Table 11).

## Volcanic gas measurements

The composition of the gas emissions (Extended Data Fig. 9) was obtained from an open-path Fourier transform infrared spectrometer and multi-component gas analyser system (Multi-GAS) measurements. We used a MIDAC Fourier transform infrared spectrometer, with liquid nitrogen cooled mercury cadmium telluride detectors and a 3-inch Newtonian telescope with 10 mrad field of view. Interferograms and single beam spectra were collected at 0.5 cm$^{-1}$ resolution approximately every 2 s. The amount of gas contributing to the measured spectra was determined using a forward model[40,41] that simulates absorptions of target volcanic and atmospheric gas molecules in a specified spectral range using line parameters taken from the HITRAN database (hitran.com). Volcanic gas composition according to the Fourier transform infrared measurements was $80.4 \pm 0.8$ mol% $H_2O$, $16.1 \pm 0.5$ mol% $CO_2$, $3.37 \pm 0.3$ mol% $SO_2$, $0.020 \pm 0.005$ mol% CO, 0.05 mol% HCl and $0.03 \pm 0.005$ mol% HF. Multi-GAS measurements[42] were conducted at source in the grounding plume and in the lofted plume. The ground Multi-GAS measures real-time (0.5 Hz) concentrations of major volcanic gas species using gas specific sensors (Gascard II, T3ST/FTD2G-1A and T3H-TC4E-1A from City Technology and KVM3/5 Galltec-Mela sensors)[43]. The Unoccupied Aerial System (model DJI Matrice 600pro) used a light, compact version of the Multi-GAS[44] with real-time measurements (1 Hz) and comparable gas specific sensors (with the same electrochemical sensors as in the ground-based system and Microsensorik Smartgas Modul Premium2 and Bluedot BME280 sensor). Initial volcanic gas composition according to the Multi-GAS measurements was $53 \pm 10$ $H_2O$, $37 \pm 9$ $CO_2$ and $11 \pm 2.5$ $SO_2$ (Supplementary Table 12).

## MI PEP correction

We determined the major element composition of a total of 200 plagioclase-, olivine- and clinopyroxene-hosted MI, in addition to the trace element and major volatile contents of 32 inclusions and 11 groundmass glasses from samples collected two days into the eruption (21 March 2021). The composition of the olivine, plagioclase and clinopyroxene host crystals was analysed next to each MI. Major elements in MIs were corrected for post-entrapment processes (PEPs) following the procedures described in ref. [45]. For trace element corrections, equilibrium plagioclase and olivine compositions were calculated from MI compositions using the partition coefficients listed in Supplementary Table 11. The calculated trace element compositions of minerals were added back to the MI composition in the amount required by PEP (Supplementary Tables 10 and 11). Most measured (groundmass glass and whole rock) and recalculated (MIs) melt compositions followed a trend enveloped by the liquid line of descent (LLD) of the enriched and depleted end-member melts of the RP[46]. Exceptions were some of the more evolved groups of olivine-hosted MIs. These inclusions were too rich in FeO and poor in CaO and $Al_2O_3$ compared to the predicted LLD by the other melt compositions. Some clinopyroxene-hosted MIs showed the same deviations from the predicted LLD (see Supplementary Information, section S4).

## Summary of methods used to obtain temperature and pressure

**Eruption temperature.** The eruption temperature was calculated based on equation (15) in ref. [47] (Supplementary Table 9). To estimate eruption

temperatures, the pressure was set to 0.001 GPa and an oxygen fugacity of FMQ-0.3 was applied. The oxygen fugacity was calculated based on equilibrated (following recommendations in ref. [48]) olivine–spinel pairs using the calibration of ref. [49]. The temperature on eruption at the surface was $1,200 \pm 20$ °C (see ref. [50] for a discussion of the uncertainty associated with the calculations) and remained the same as the eruption progressed over the first six weeks of the eruption, regardless of the change in magma composition.

**Melt barometry.** The last equilibration pressure of the magma and its crystal cargo before the eruption was calculated using the olivine-plagioclase-augite-melt (OPAM) (ref. [20] and references therein) and clinopyroxene–melt[51] geobarometers (Supplementary Tables 5, 9 and 10). A Jupyter Notebook is supplied in the Supplementary Information with the scripts used to perform the calculations. Extended Data Figure 6 shows the relationship between the calculated pressures and temperatures; in this figure, for OPAM barometry, the results from Supplementary Tables 9 and 10 are only plotted when the probability fit of the melt composition for the four-phase cotectic[20] is over 0.8. The calibration uncertainty of both geobarometers applied is ±1.3–1.4 kbar. MI temperatures were calculated by using equation (15) of ref. [50], with input pressures obtained from OPAM barometry. The most primitive MI, found in olivine with a Fo content of up to 89, gave temperatures of more than 1,300 °C.

**Clinopyroxene barometry.** For this, we used two different approaches. The first approach was the 'statistical approach' of ref. [51]. A total of 354 clinopyroxene point analyses were carried out in 131 grains. These were fitted with putative equilibrium liquid compositions and tested with equilibrium criteria. We evaluated the equilibrium conditions by matching each clinopyroxene composition with equilibrium liquid compositions ($n = 283$) consisting of groundmass glass, PEP-corrected MIs and whole-rock compositions. The equilibrium criteria were as follows: (1) Fe–Mg partitioning between clinopyroxene and melt, $Kd_{Fe-Mg}^{cpx-melt} = 0.27 \pm 0.03$ (ref. [47]); (2) Ti partitioning ($D_{Ti}$) between clinopyroxene and melt using the lattice strain model of ref. [52], and selected clinopyroxene–melt pairs that were within 40% of the predicted Ti equilibrium; (3) clinopyroxene–melt pairs with predicted and observed DiHd, EnFs and CaTs clinopyroxene components[53] lower than ±0.06, 0.05 and 0.03, respectively[54]. In total, 202 clinopyroxene compositions were successfully matched with a putative equilibrium liquid composition (Supplementary Table 5). A second approach was to calculate pressure based on clinopyroxene–MI pairs when it was possible. Here we used the PEP-corrected MI compositions and calculated the pressures with the host pyroxene of the inclusions. The melt–clinopyroxene pairs still had to fulfil the same equilibrium criteria as in the statistical approach. Supplementary Table 5 shows the comparison of the results for those grains for which both approaches were applicable. Both pressures and temperatures (based on equation (33) in ref. [50]) show an excellent fit between the two methods within the uncertainty of the calibrations.

**Saturation pressures from $CO_2$ and $H_2O$ in MIs.** The PEP-corrected MIs record $H_2O$ contents of 0.18–0.27 wt%, S contents of 530–1,026 parts per million weight (ppmw, or 1,059–2,053 ppmw $SO_2$), and $CO_2$ concentrations ranging from 87 ppmw to 2,136 ppmw $CO_2$ (Extended Data Fig. 7). The pressures at which a $CO_2$–$H_2O$ fluid would be saturated for each MI (both with the raw and PEP-corrected major and volatile element concentrations) were calculated with the MagmaSat[55], Iacono-Marziano[56] and Shishkina[57] models, using v.1.0.1 of the VESIcal software[56]. The $Fe^{3+}/Fe_T$ ratio was assumed to be 0.15 in all inclusions, and the temperature was set to 1,200 °C, but the models showed little sensitivity to these parameters. A Jupyter Notebook used to perform the calculations is provided in the Supplementary Information. Saturation pressures calculated for the most $CO_2$-rich inclusions are between 0.3 and 0.4 GPa; however, they have most probably been affected by host decrepitation during

ascent[58,59], and so provide only a minimum bound on the pressure of magma storage (Extended Data Fig. 8).

Propagating analytical uncertainty in $CO_2$ concentration to saturation pressure yields a $1\sigma$ error of less than 7% in all but the lowest saturation pressures. This uncertainty is significantly smaller than the uncertainty in model calibration[60], as indicated by the discrepancies between saturation pressures calculated using the models in refs. [55–57] (Extended Data Fig. 8). As a conservative representation of this uncertainty, we use the highest saturation pressure calculated with the Shishkina model[57] as a minimum bound on the inclusion entrapment pressure.

**Surface gas emissions.** The $CO_2/S$ mass ratios preserved in the inclusions (0.2–2.6) are considerably lower than those observed in the surface vent gas emissions (5.2 ± 1.3; Extended Data Figs. 7 and 9). This indicates that the parental magmas, and the inclusions themselves, have lost a significant proportion of their $CO_2$ to the gas phase during storage and ascent, as seen ubiquitously in other volcanic systems[59]. The open-path Fourier transform infrared spectrometer/Multi-GAS dataset (Supplementary Table 12) constrain the 'average' volcanic gas $CO_2/S$ (mass) ratio at 5.2 ± 1.3. This ratio, in combination with trace element whole-rock composition (Supplementary Table 3), place Fagradalsfjall well within the global compositional array[61] of plume-related (MOR and intraplate) and continental rift volcanism (Extended Data Fig. 10). Such a correlation with ratios between incompatible/non-volatile trace elements (for example, Sr/Sm and Sr/Nd ratios), which are unaffected by magma degassing upon ascent and decompression, has been taken[61] as evidence that the volcanic gas $CO_2/S$ ratios are controlled by source mantle characteristics, so, for example, the degassing-driven fractionation has little control on the gas signature. This can only be obtained if a closed-system degassing behaviour has prevailed until shallow magmatic levels, which is a condition typically met at mafic volcanoes[62]. The corollary is that the Fagradalsfjall volcanic gas $CO_2/S$ ratio can be used, in combination with the difference in S contents between the MIs and G20210428 fountain tephra (1,490 ± 350 ppmw), to infer a pre-eruptive dissolved $CO_2$ concentration of 3,880 ± 1,030 ppmw. This concentration places a minimum bound on the pressure at which the magma was stored, assuming the gas phase was produced on ascent: the $CO_2$ vapour saturation pressure. As above, the saturation pressure (and its uncertainty) was calculated using the Shishkina model[57] with VESIcal software[63] to be 0.49 ± 0.11 GPa (18 ± 4 km); the MagmaSat[55] and Iacono-Marziano[56] models returned higher saturation pressures (0.67 ± 0.15 GPa and 0.59 ± 0.10 GPa, respectively). In each calculation, the bulk composition was set to the average glass composition of fountain tephra G20210428 and the $Fe^{3+}/Fe^T$ value was set to 0.15, at a temperature of 1,200 °C. This calculation yielded a $CO_2$ saturation pressure of 0.51 ± 0.15 GPa. A Jupyter Notebook is supplied in the Supplementary Information with the scripts used to perform the calculations.

## Data availability

Source data are provided with this paper for all figures and also in the Supplementary Information and Tables. A Jupyter Notebook is also supplied in the Supplementary Information. The data are also available at EarthChem (https://doi.org/10.26022/IEDA/112319).

## Code availability

Version 1.0.1 of the VESIcal software used for the $CO_2$–$H_2O$ solubility calculations is available on Zenodo (https://doi.org/10.5281/zenodo.5095382). The scripts and data files used to run VESIcal and alphaMELTS, as well as the script for performing the OPAM geobarometry calculations, are also available on Zenodo (https://doi.org/10.5281/zenodo.6631329).

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

**Acknowledgements** We are grateful for the analytical help of C. Bosq and D. Auclair during the Sr and Nd isotope measurements. We thank G. Pedersen for providing shape-files of the lavas and M. T. Guðmundsson, H. Geirsson, F. Sigmundsson, B. Brandsdóttir, P. Einarsson, K. Grönvold, N. Óskarsson and K. Sæmundsson for helpful discussions and support. We thank S. Johnson, H. Rúnarsdóttir and Á. Ásmundsdóttir for assistance with sample preparation. The NordSIMS ion microprobe facility acknowledges support by the Swedish Research Council (grant no. 2017-00671), the Swedish Museum of Natural History and the University of Iceland; this is NordSIMS publication no. 713. The involvement of S.A.H. was partly in relation to H2020 project EUROVOLC, funded by the European Commission (grant no. 731070). This work was supported by the Icelandic Research Fund, grant no. 228933-051. We gratefully acknowledge constructive comments provided by A. Kent and K. Rubin that helped to improve this work. A.A. ackowledges funding from Italian Ministero Istruzione Università e Ricerca (Miur), grant PRIN2017-2017LMNLAW.

**Author contributions** E.W.M., A.C., S.M., M.B.R., E.R., G.K.M. and M.H.K. collected samples. E.W.M., A.C., S.M., E.B., M.B.R., E.R., J.G.R., G.H.G., O.S., M.J.W., H.J., Q.H.A.v.d.M., G.K.M., M.H.K., M.M.R., R.H.R. and G.S. prepared and analysed all rocks samples. M.A.P., S.W.S., R.K., B.I.K., C.O., A.A., E.I., M.B., G.G. and A.S. collected and analysed all gas data. S.A.H. wrote the first version of the manuscript with critical input from E.W.M., A.C., S.M., E.B., G.H.G., O.S., J.M., M.G.J. and A.S. All authors contributed to data interpretations, critical discussions and commented on the manuscript.

**Competing interests** The authors declare no competing interests.

## Additional information

**Correspondence and requests for materials** should be addressed to Sæmundur A. Halldórsson.

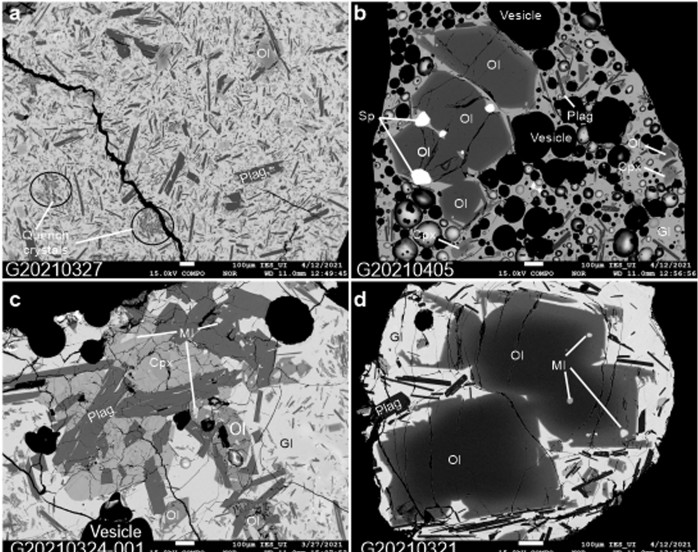

**Extended Data Fig. 1 | Petrographic features of the Fagradalsfjall lava.** Petrographic features of the Fagradalsfjall samples seen in backscattered electron images. a. Quenched lava, essentially without macrocrysts. b. Tephra with olivine macrocrysts containing spinel inclusions. c. Glomerocryst of Plag-Cpx-Ol with MI (important for OPAM). d. Macrocryst (Ol) with MIs in both core and rim (rim in equilibrium with the carrier melt, core is too primitive). Ol – olivine, Plag – plagioclase, Cpx-clinopyroxene, Gl – silicate glass, Sp – Cr-spinel, MI – silicate melt inclusion.

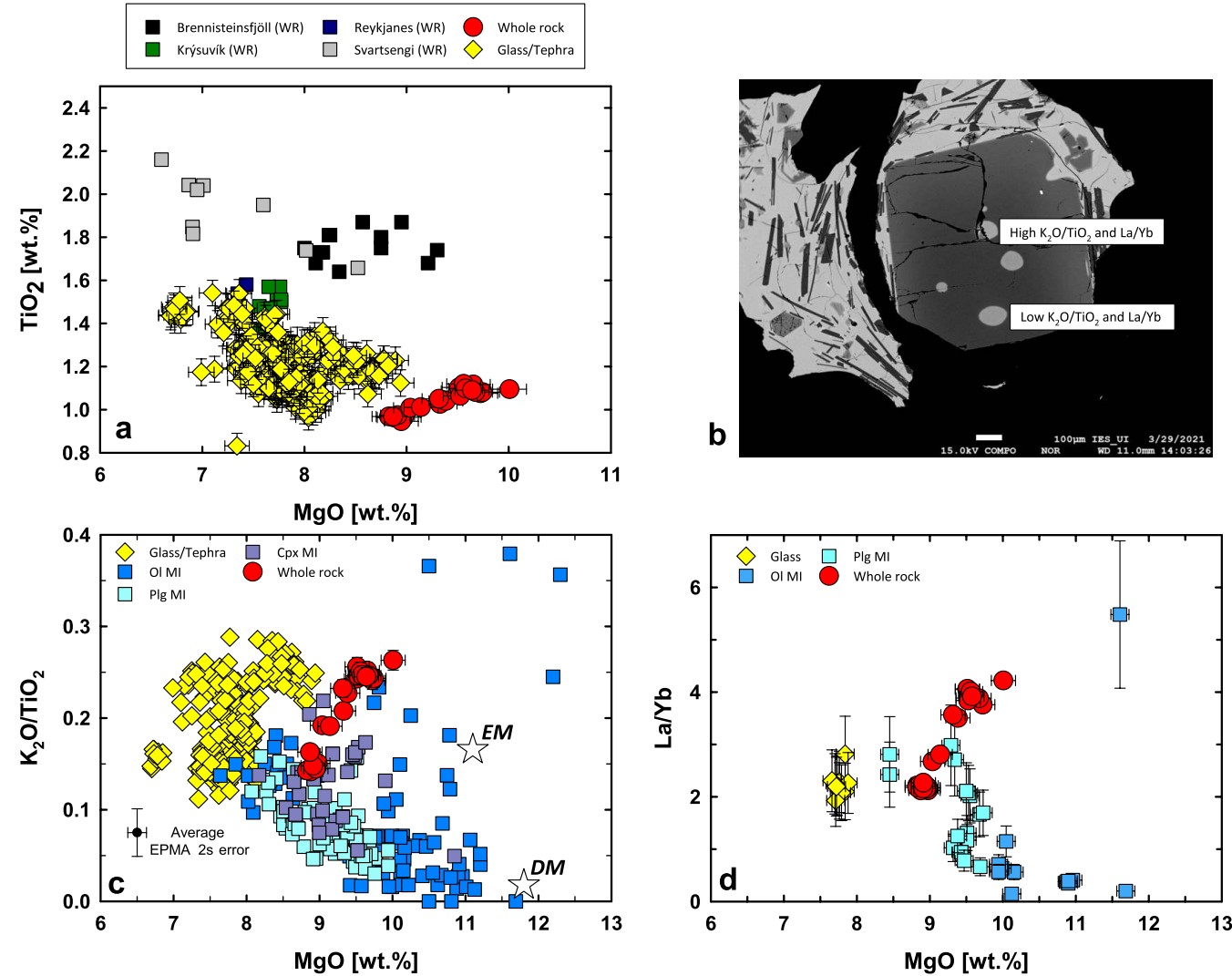

**Extended Data Fig. 2 | Chemical Characteristics of the Fagradalsfjall lava.** a, TiO$_2$ versus MgO for whole rock and glass/tephra samples of the Fagradalsfjall eruption. For comparison, whole rock samples from other intra-transform spreading centers on the Reykjanes Peninsula are also shown[14,17,18]. b, A single olivine crystal from sample G20210321-4 showing MIs with both low and high K$_2$O/TiO$_2$ and La/Yb. c-d, K$_2$O/TiO$_2$ and La/Yb vs. MgO, for whole rock and tephra samples compared to plagioclase (Plg), olivine (Ol) and clinopyoxene (Cpx) melt inclusions. In c, white stars represent the depleted (DM) and enriched (EM) parental melt end-member of the Reykjanes Peninsula[21]. Error bars are included on all panels and include external 2σ error for all whole rock and glass data (where average EPMA 2σ error is reported in the lower left corner) but 1σ error the SIMS data plotted in panel d.

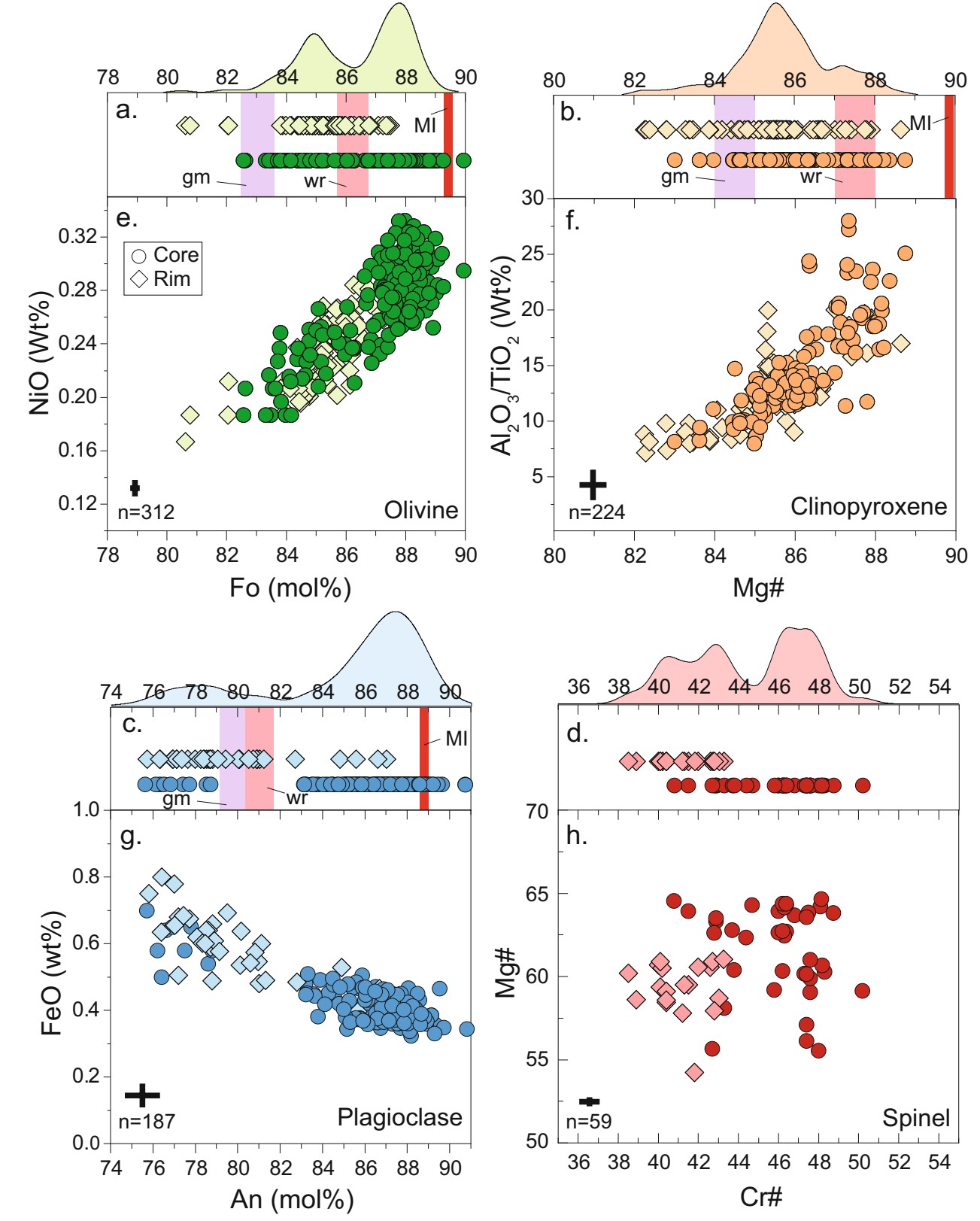

**Extended Data Fig. 3 | Mineral compositions.** The range in a. Fo content of olivine, b. Mg# of clinopyroxene, c. An content of plagioclase and, d. Cr# of spinel. Core and rim compositions are depicted as circles and diamonds, respectively. Red bar shows the mineral compositions in equilibrium with the most primitive melt inclusions, pink bars are equilibrium compositions with a whole rock-like liquid, whereas purple bars indicate mineral compositions in equilibrium with tephra glass sampled in March. Numbers in each corner state the number of point analyses in minerals. Variation diagrams showing the NiO vs Fo content of olivine e., Al$_2$O$_3$/TiO$_2$ vs Mg# clinopyroxene f., FeO content vs An content of plagioclase g. and Mg# vs Cr# in spinel macrocrysts h. Kernel density estimates on the top axis produced using a bandwidth of 0.2 show the relative probability of mineral compositions, determining the main compositional populations in each mineral. Representative error bars are included on all panels and include external 2σ error.

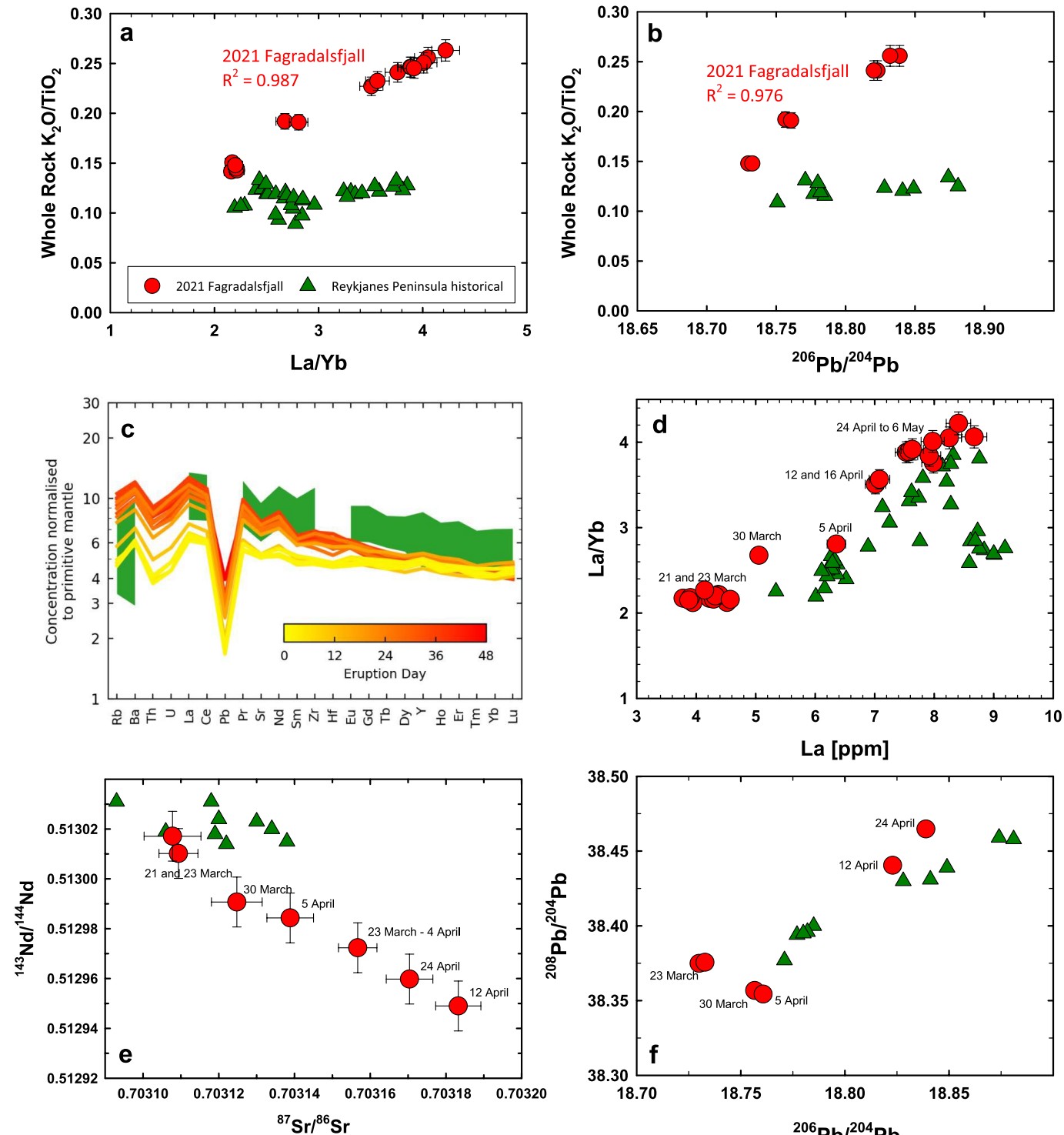

**Extended Data Fig. 4 | Trace elements and radiogenic isotope compositions.** a, La/Yb vs. $K_2O/TiO_2$. b, Pb isotope vs $K_2O/TiO_2$, c, Primitive mantle normalized trace element patterns for the Fagradalsfjall samples reported here. Normalised to the primitive mantle[62]. Lighter colors represent earlier eruption dates. d, La vs. La/Yb with samples dates. e, Sr vs. Nd isotopes plotted with samples dates. f, Pb isotope plot with samples dates. Comparative data from other Reykjanes Peninsula lavas erupted historically (i.e., erupted after Settlement, circa AD 870) are from[14,17,18]. Error bars are included on panels except c and include external 2σ error.

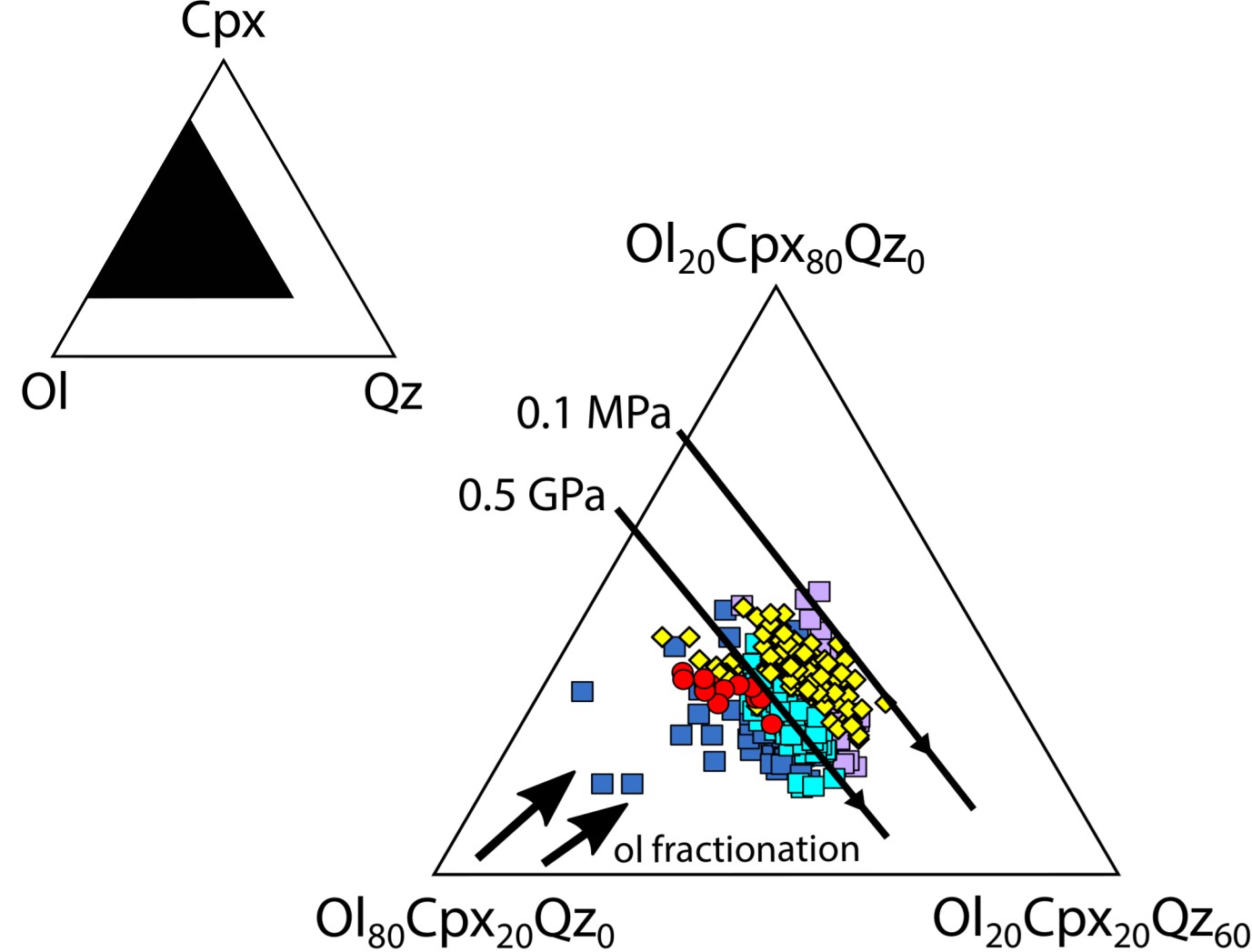

**Extended Data Fig. 5 | CIPW molecular normative plot of glass, melt inclusions and whole rock compositions.** CIPW molecular normative plot, which includes the tholeiitic portion of the basalt tetrahedron, Ol-Cpx-Plag-Qz, projected from the plagioclase apex[64]. The approximate location of the cotectic with olivine (ol), plagioclase (plag) and clinopyroxene (cpx) in equilibrium with melts like those found in the Fagradalsfjall eruption products is shown for two different pressures, 0.1 MPa and 0.5 GPa (calculated using the OPAM code in ref. [20]). The compositions of the most primitive olivine MI are consistent with a short temperature interval of olivine-only fractionation. At a lower temperature, plagioclase and clinopyroxene join olivine as crystallizing phases. Symbols are the same as shown in Extended Data Fig. 2c, d.

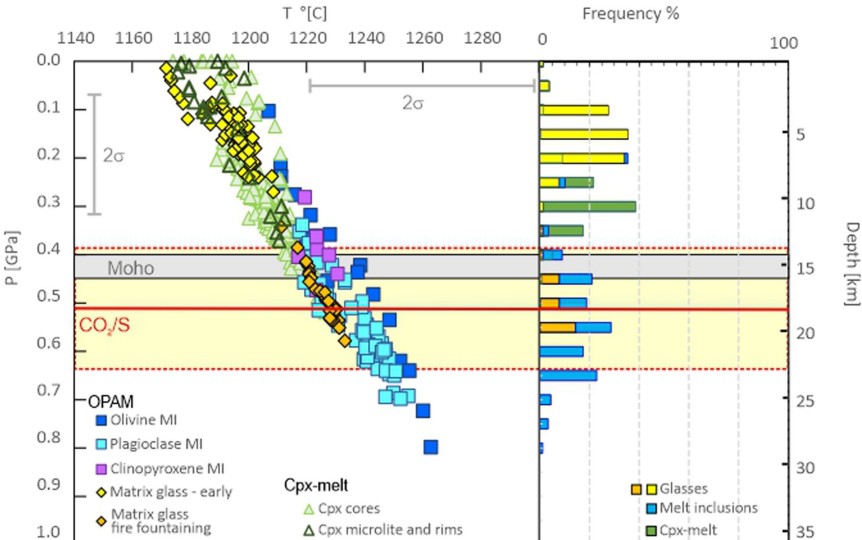

**Extended Data Fig. 6 | Inferred pressures and temperatures of crystallization from thermobarometry.** Using the olivine–plagioclase–augite–melt (OPAM) barometer (Methods), we calculate that the carrier melt of the crystal cargo, inferred from groundmass glass compositions equilibrated over the pressure range of 0.05 to 0.25 GPa in samples collected from March and early April. This coincides with calculated crystallization pressure of the evolved clinopyroxene macrocryst rims and cores suggesting crystallization in the mid- to upper crust. In contrast, glasses from the fire fountaining phase (late April-early May) and MI equilibrated at much higher pressures with the primitive crystal cargo, from 0.36 to 0.8 GPa with most probable pressures of 0.55 to 0.65 GPa. This is close to the values obtained from the cores of some of the primitive clinopyroxene macrocrysts (up to 0.52 GPa –Methods). We have also calculated the expected $CO_2/SO_2$ in the gas phase, assuming $CO_2$ vapour saturation at various pressures of storage, followed by near-complete degassing of $CO_2$ and $SO_2$ during eruption (Methods). The contour lines show that the $CO_2/SO_2$ ratio observed at the vent can only be obtained if the magma rises from at least 0.5 GPa pressure. Model error ($2\sigma$) is indicated with the grey error bar. See Methods for details.

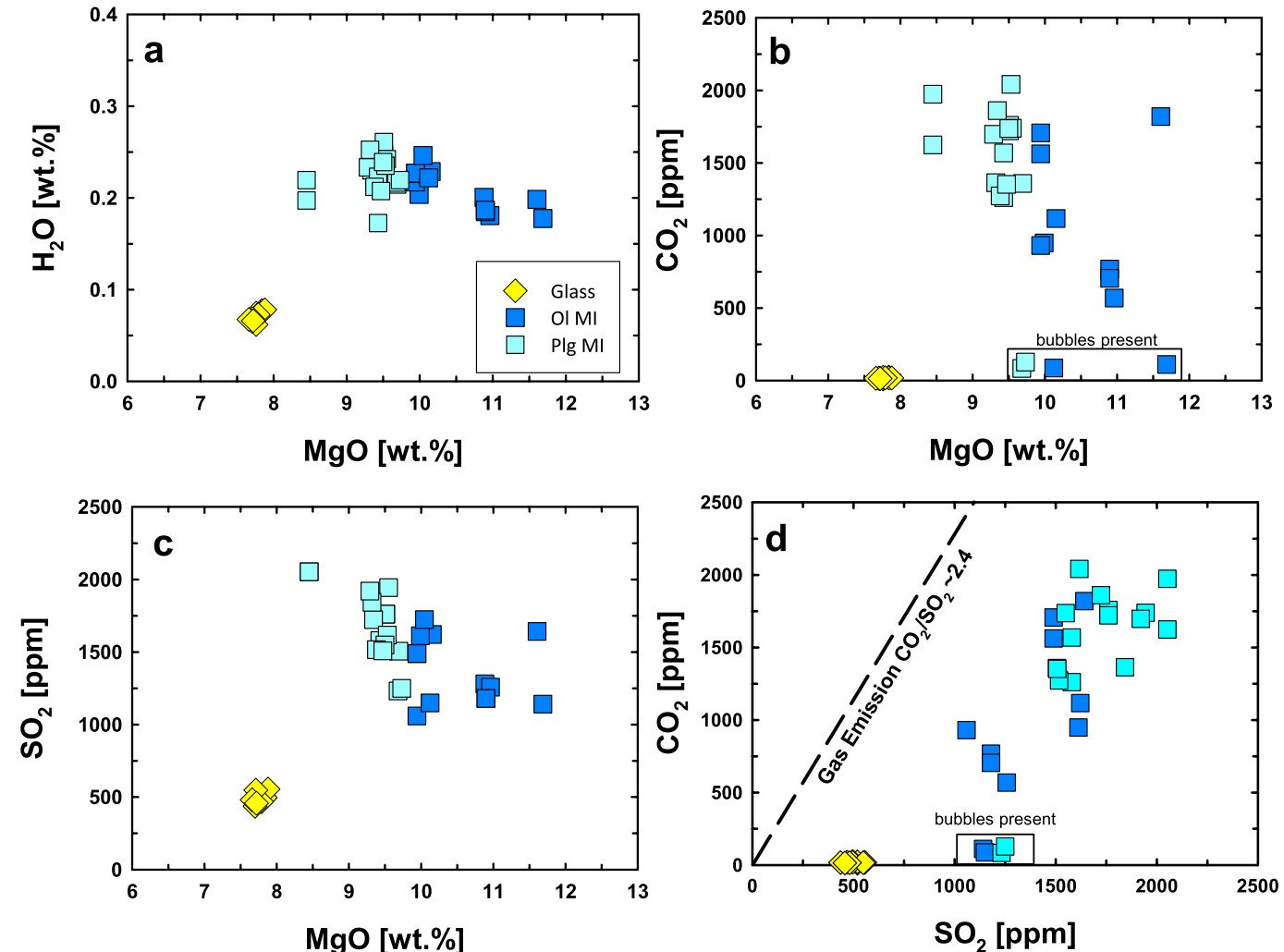

**Extended Data Fig. 7 | Volatile elements in melt inclusions and comparison with gas emissions.** a and c, The PEP-corrected olivine- and plagioclase-hosted MI record overlapping $H_2O$ (0.18 –0.26 wt.%) and $SO_2$ (1059–2053 ppm) contents. b, In contrast, bubble-free olivine-hosted inclusions have lower median $CO_2$ contents (1126 ppm) than the bubble-free plagioclase-hosted inclusions (1649 ppm). d, Volatile elements in melt inclusions and comparison with gas emissions. In b. and d. samples that contain gas bubbles are indicated. Error bars are included on all panels and include external 1σ error. In all panels, are the 1σ error bars smaller than the symbol size.

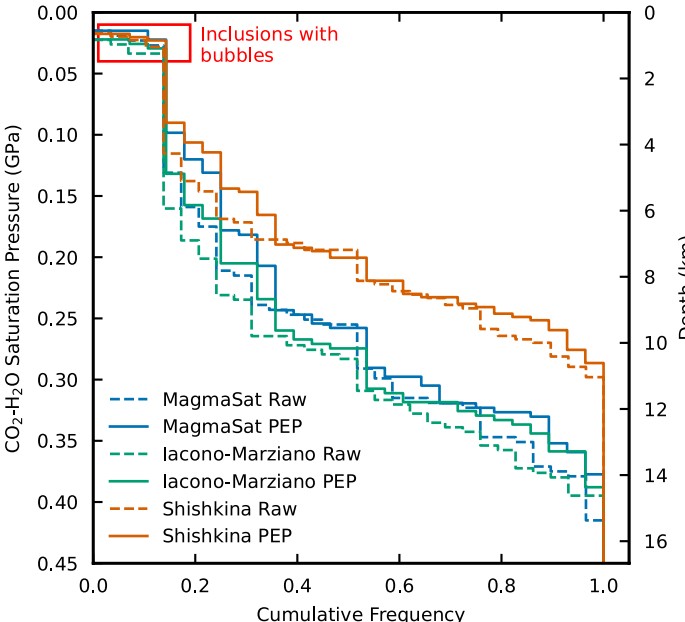

**Extended Data Fig. 8 | Cumulative frequency of melt inclusion $CO_2$-$H_2O$ saturation pressures.** The $CO_2$-$H_2O$ saturation pressures calculated for the olivine and plagioclase hosted MI, with and without PEP corrections. Calculations performed using v1.0.1 of VESIcal[63] at 1200°C, assuming $Fe^{3+}/Fe_T = 0.15$ with the MagmaSat model[55], the Iacono-Marziano model[56], and the Shishkina model. See Methods for discussion on the model uncertainty.

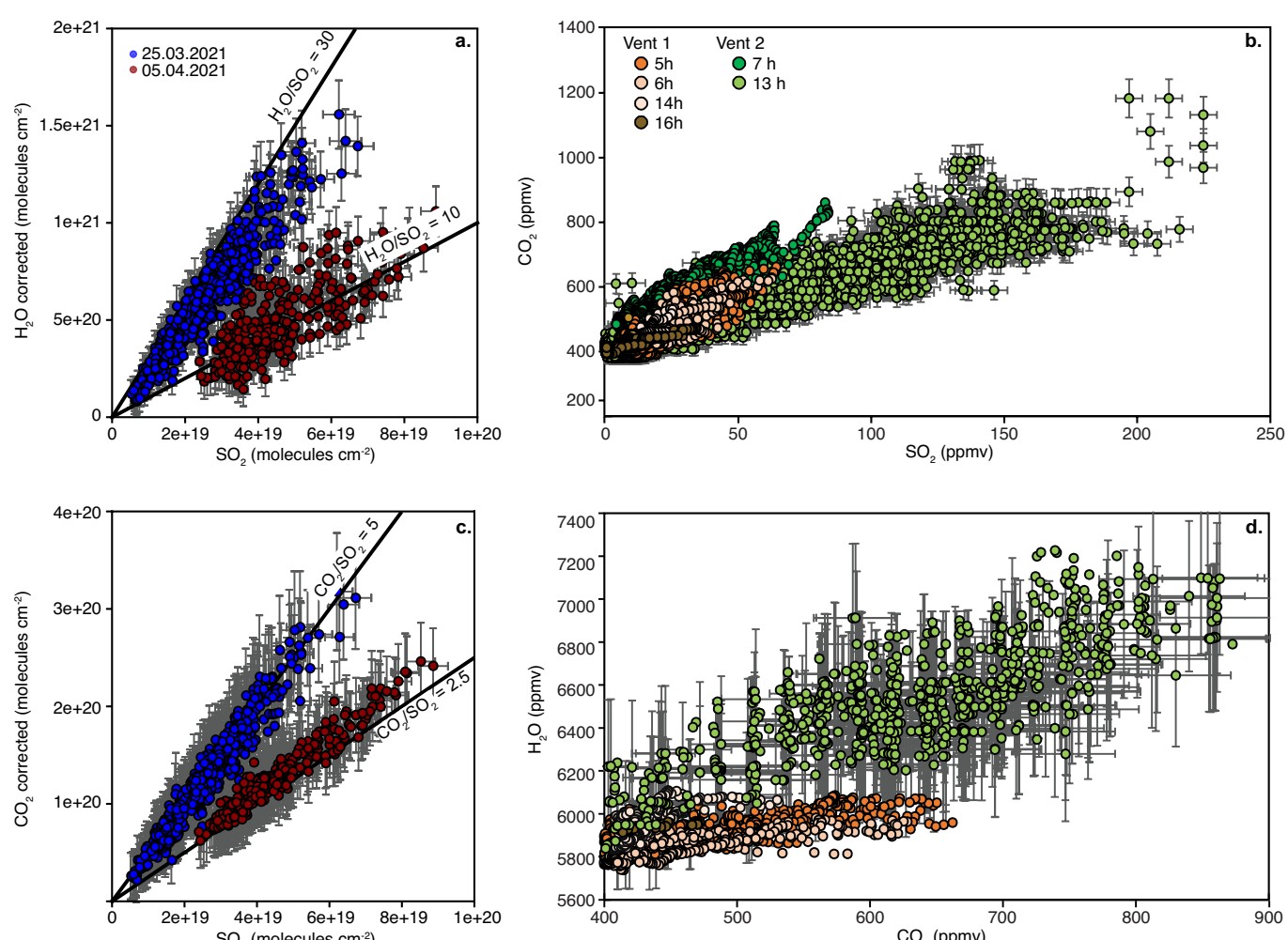

**Extended Data Fig. 9 | Characteristics of gas emissions.** a-b: Plot of (a) $H_2O$ vs $SO_2$ and (b) $CO_2$ vs. $SO_2$ column amounts for vent gases from fracture one during 25th March and 5th April. The measurements were carried out using an open-path Fourier transform infrared spectrometer (OP-FTS) at approx. 150 m distance. Note that atmospheric corrections have been applied to the $H_2O$ and

$CO_2$ measurements. c-d: Plot of (c) $CO_2$ vs. $SO_2$ and (d) $H_2O$ vs $CO_2$ for vent gases on the 25th March and 5th April. The measurements were carried out using a Multi-GAS device at 50-150 m from the vent source. Error bars are included on all plots and include external 2σ error.

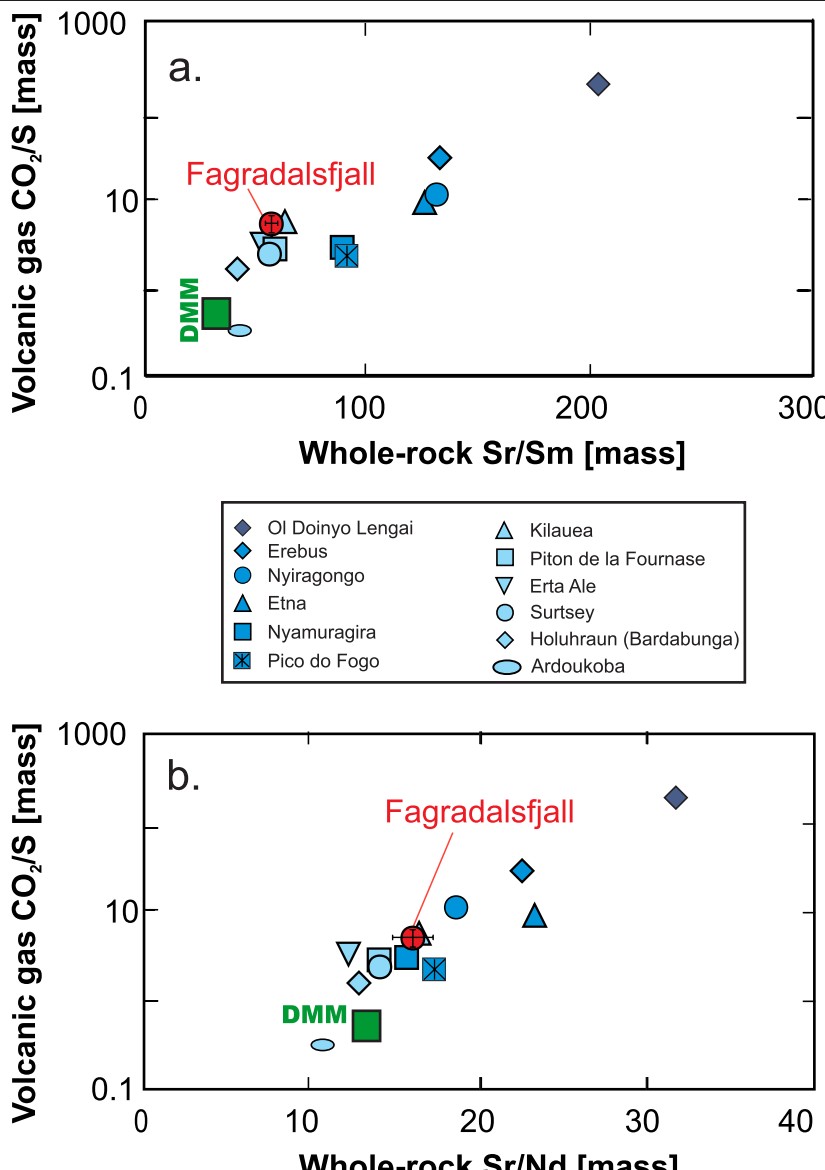

**Extended Data Fig. 10 | The Fagradalsfjall gas signature in a global context.** Time-averaged volcanic gas $CO_2$/S ratios versus mean whole-rock Sr/Sm and Sr/Nd ratios for a global population of plume-related (MOR and intraplate) and continental rift volcanoes[61], and for Fagradalsfjall (this study; Supplementary Tables 3 and 12). The gas/trace element composition of the Depleted MORB Mantle (DMM) is from ref.[61]. These correlations are evidence for that closed-system degassing behavior has prevailed until shallow magmatic levels, and that the volcanic gas $CO_2$/S ratio can thus be used to infer the parental melt $CO_2$ concentration.

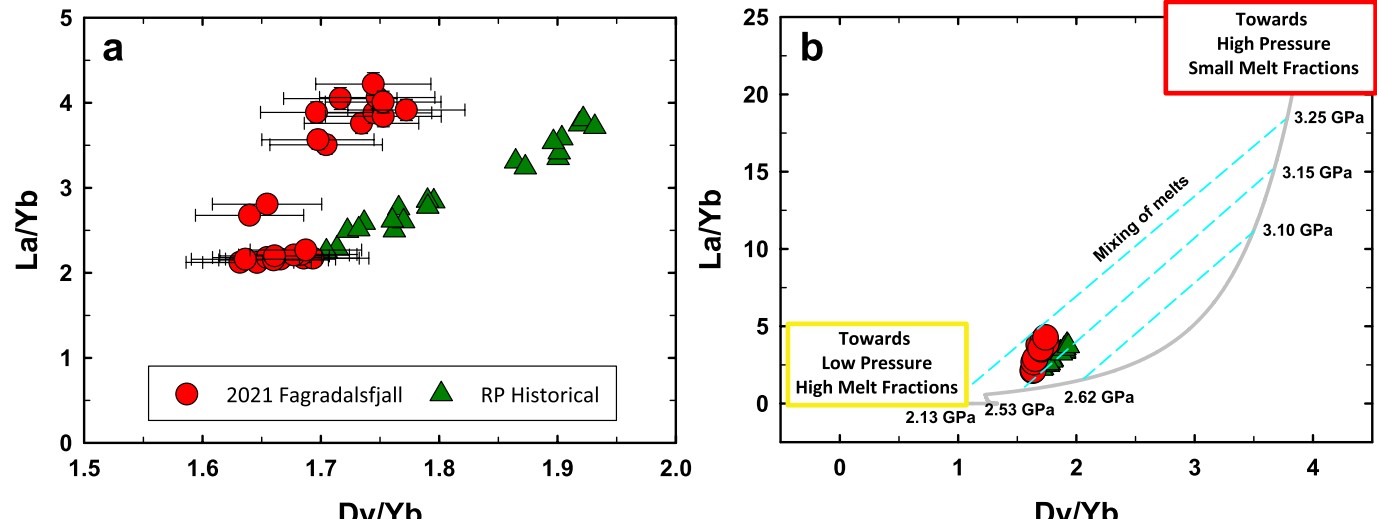

**Extended Data Fig. 11 | REE systematics show variability derived from mantle melting.** a, Dy/Yb vs. La/Yb of the Fagradalsfjall whole rocks, compared with historial RP eruptions. b, the same data superimposed onto a mantle melting trajectory, with indicative mixing lines. The calculation assumes a homogenous depleted mantle composition[65] and was performed using alphaMELTS[66] running the pMELTS model[67]. The calculations were for isentropic decompression melting starting at 4 GPa and 1572 °C, with 0.01 GPa pressure steps. Trace element concentrations were calculated using constant partition coefficients from ref. [68,69], and assuming continuous melting with a residual porosity of 0.2%. The pressures indicated are model dependent. Error bars are included on both panels and include external 2σ error.

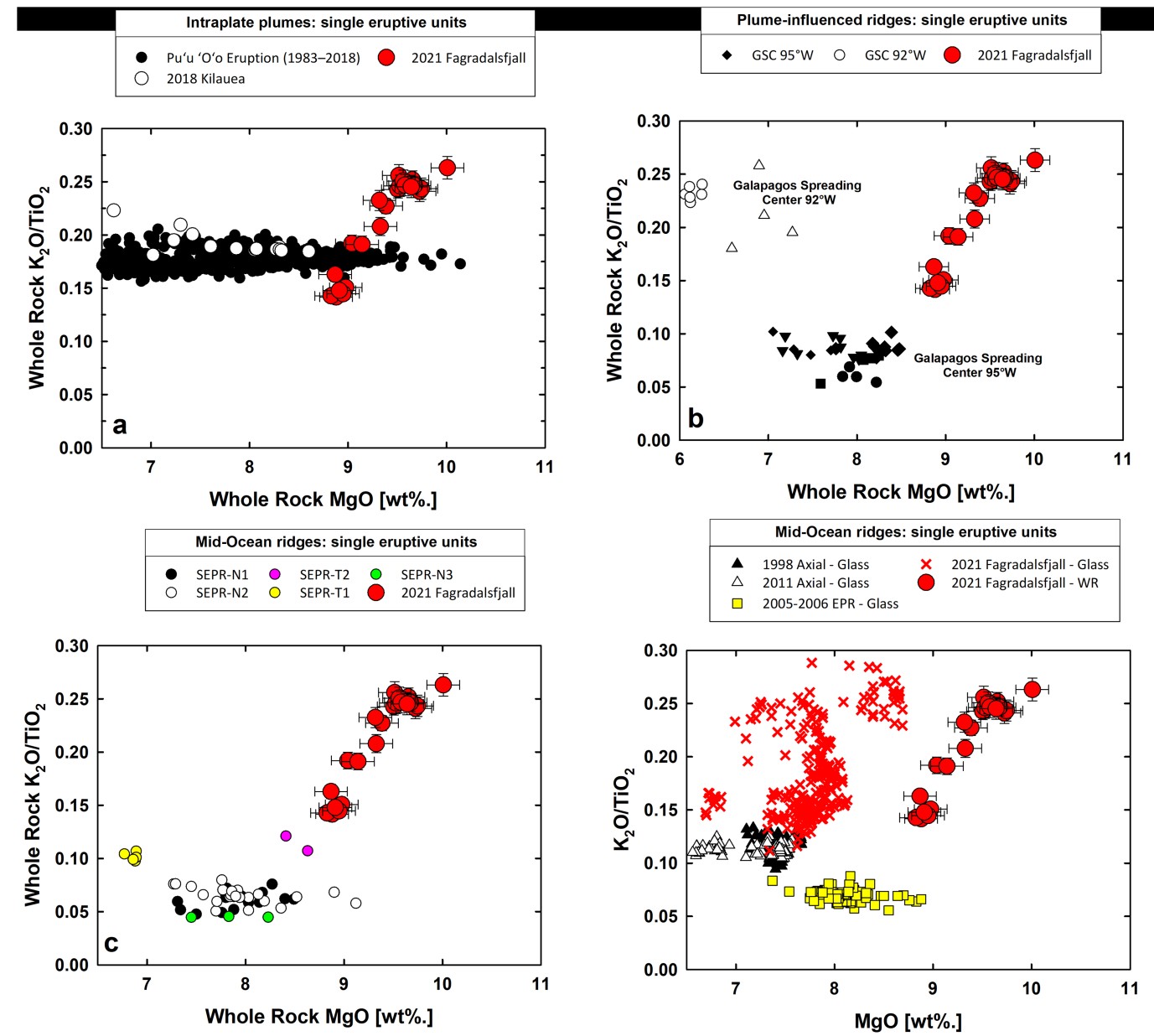

**Extended Data Fig. 12 | Fagradalsfjall K$_2$O/TiO$_2$ variability vs. single-eruptive oceanic basalt units.** Comparison of the Fagradalsfjall K$_2$O/TiO$_2$ variability with petrologically well-characterized single-eruptive oceanic basalt units that have been mapped and sampled for within-flow variations. a, Pu'u 'Ō'ō 1983–2018 and Kilauea 2018 eruption[29,31] and individual MOR basaltic eruptions from b, plume-influenced ridges along the Galápagos spreading center[25], where different units are shown using different symbols. c, units near 17.5°S on the East Pacific Rise (EPR)[26], and d, some newly-formed EPR (2005–2006) eruptions[27] and recent eruptions on the Axial seamount[28]. Except for the Galápagos spreading center[25], only lavas flows with MgO contents higher than 6.5 wt.% are shown. Error bars are included on all panels and include external 2σ error.

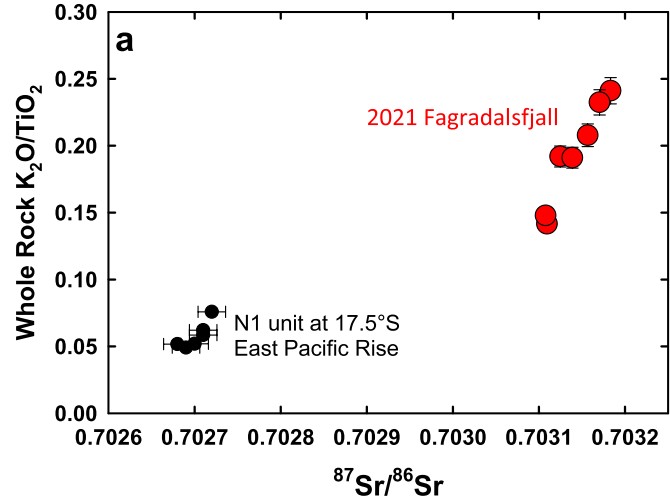 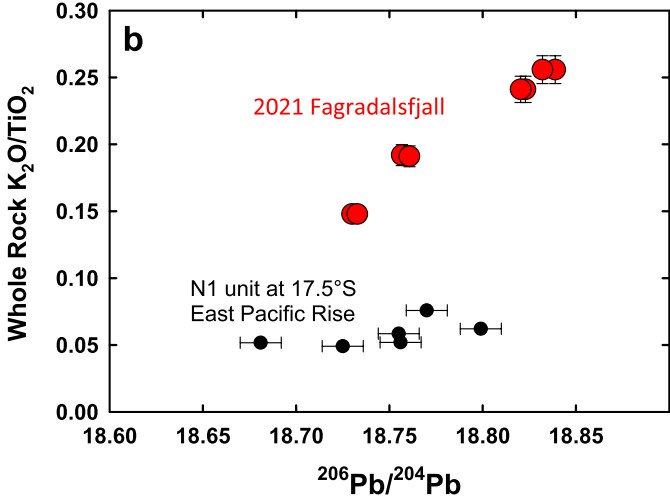

**Extended Data Fig. 13 | Fagradalsfjall K₂O/TiO₂ vs. the N1 lava unit at 17.5°S on the East Pacific Rise.** a and b, Comparison of the Fagradalsfjall K$_2$O/TiO$_2$ with the N1 lava unit at 17.5°S on the East Pacific Rise which shows both considerable and systematic mantle-derived heterogeneity[26]. Error bars are included on both panels and include external 2σ error.