## [Peer Review File · Nature]

Manuscript Title: Rapid shifting of a deep magmatic source at Fagradalsfjall volcano, Iceland

Reviewer Comments & Author Rebuttals

Reviewer Reports on the Initial Version:

Referee #1 (Adam Kent):

Review of “Rapid evolution of a deep magmatic system revealed by the Fagradalsfjall eruption, Iceland” by Halldórsson et al.

The manuscript reports a large data set obtained from geochemical, petrological and gas measurements from the recent Fagradalsfjall eruption on the Reykjanes Peninsula of Iceland. The paper presents data for this eruption that will be tremendously valuable for future workers in a variety of areas – including volcanology, mantle geochemistry, igneous petrology, and hazards. The work is also certainly novel as this is an exceptional eruption.

The manuscript makes the reasonable case that the eruption at Fagradalsfjall is different to other recent rift eruptions in Iceland and elsewhere, in that there is little evidence for shallow crustal storage, with magmas coming directly from the upper mantle or lower crust. This means that there is lesser crustal processing, and thus the opportunity to see mantle processes and mantle derived compositions with more fidelity. Geochemical variations also suggest that there was mixing between magmas derived from distinct enriched and depleted mantle domains and a rapid shift between these compositions during the ongoing eruption. Although the recognition of rapid transport of magmatic components from the mantle or lower crust is not completely new, and rapid transport of phenocrysts from the mantle or lower crust to the surface has been documented elsewhere in Iceland (see for example the work of Much et al. *Nature Geosciences* 2019), the opportunity to observe and fully document an eruption of this type is certainly a major opportunity – particularly for understanding the ultimate control that mantle processes might have on eruptions.

My major comments on the manuscript are below, followed by some more minor issues.

Synthesis: I very much appreciate the uniqueness of this eruption, the size and quality of the data sets presented, and there is significant opportunity for learning something new, but I also had trouble seeing what were the major scientific insights that stem from these observations and data. As such this paper reads more to me as an exceptional documentation of this eruption, but less synthesis of what it tells us that is new (or confirms other existing ideas) about ridge systems. With some quibbles (see below) I broadly accept that the erupted magmas come from the upper mantle or lower crust, with little or no shallow crustal processing, but I do not feel like I learned something new about how mid ocean ridge magma systems work. Given the importance of the paradigm of the melt lens and shallow processing for many magmas in mid ocean ridges show I feel it is reasonable to expect a greater level of synthesis here.

Interpretation as liquid: One important issue that needs clarification is the interpretation of the

whole rock compositions as a liquid, which happens both explicitly and implicitly throughout the manuscript. This assumption is difficult to reconcile with the observation of abundant mafic and zoned phenocrysts, and many mafic magmas are mixtures of evolved liquids and antecrysts. I was surprised that there is no clear discussion of this important issue, and without that I have some trouble fully accepting some of the barometry and other key parts of the interpretation. One essential piece of information lacking in this respect is the modal proportions of crystals that are present and if these proportions vary through time. It is essential to provide these – and if possible, give the proportions of different crystal types identified. Variations in whole rock chemistry may well reflect variations in modal crystal proportions through time, rather than changes in some primitive igneous liquid.

Barometry: The barometry is an important part of the interpretation that magmas do not experience shallow crustal residence, however there are some limitations with respect to the data supplied that prevent a full assessment of this interpretation, mostly revolving around assessing uncertainties. There is increasing awareness that uncertainties in calculated pressures from barometry can be much larger than the canonical uncertainties used here (such as the 1.4 kbar uncertainties for cpx-liquid from the original work of Putirka - e.g. see Wieser et al. 2021 – GSA Meeting abstract) as these do not fully take into account propagation of measurement uncertainties, differences in electron probe calibrations between laboratories, as well as the uncertainty derived from the calibration itself. Although it is perhaps unfair to request that these authors hold themselves to a higher standard than most other currently published work, at the very least they should provide enough metadata that these uncertainties can be independently evaluated, and there is a general level of incompleteness and inconsistency with the supplied metadata and uncertainties (see below). For example the calculated pressures for melt inclusions should also include uncertainties related to calibration and measurements (or be specific about exactly what these uncertainties represent). I am less familiar with the OPAM barometer but I also wonder what measurement uncertainties do when propagated through this calculation. In addition, all code used to determine pressures should also be released (I believe this is nature policy). I am not aware that the OPAM code is publicly available. I am also confused why the pressures of melt inclusions are not included in Figure 4 as they should also be useful for this interpretation. Finally, I also note inconsistency in the units used for pressure – where between text and figures kbar, bar and GPa are all used (I recommend using GPa as it is an SI unit).

Metadata: As noted above there is a general incompleteness and inconsistency in the metadata supplied to support the various data sets. This includes details on calibration standards, count times and secondary standards. I recommend including the following:

1. A list of actual calibration standards used for each element and count times for every element in all electron probe measurements (these are noted as being in Table 1-5 in the text but I could not find them).
2. Reported uncertainties for every element in every electron probe analysis (as provided for LA-ICP-MS and SIMS analyses for example).
3. Complete analyses of all materials analyzed as secondary standards for all analyses. At present there are no such data that I can see.
4. A clear indication how uncertainties provided for SIMS and LA-ICP-MS were determined and a clearly labelled and consistent use of 1s or 2s for ALL uncertainties in text and figures.

5. There is also very inconsistent use of error bars to reflect uncertainties in figures. Some figures have errors but not for all data, some have no error bars at all. Please include a representation of uncertainties in all figures where data is plotted, either as individual error bars or representative error bars.

Other comments linked to line numbers:

Line 113. It looks like K_2O/TiO_2 correlates broadly with isotopic composition – if so, how is sensitive to degree of melting? I suspect it is difficult to deconvolve degree of melting and source variations here.

Line 148 and elsewhere. It would be good to understand the uncertainties associated with the corrections for post entrapment crystallization (PEC) and also the sensitivity of elements to this correction. For example, MgO in olivine hosted melt inclusions is very sensitive to PEC (Kent 2008 RiMG vol 69), and even small variations in olivine addition produce large MgO variations. Using corrected compositions for further calculations – such as thermobarometry – could also be associated with large uncertainties in P and T, so this issue should be discussed.

Line 162. This is a good place to note that saturation pressures only reflect bubble free melt inclusions, as this is important.

Line 221. Although the range of compositions observed is large, it is also important to note that you do not get to equivalently sample all stages of earlier Reykjanes Peninsula eruptions, and more variable compositions could be obscured by being covered by later erupted material. What range of samples would you get if you just were able sample material present only at the later stages of the eruption?

Line 538. This additional correction sounds like it would introduce further uncertainty – has this been included in the reported uncertainties?

Line 596. Note that TiO_2 will be impacted by post entrapment crystallization in that it will be increased in concentration (as will all incompatible elements) by crystallization.

Line 629. Please provide a source for these quoted uncertainties and state which uncertainty components that they include.

Line 706. Why are glass analyses only shown for K_2O/TiO_2 but not the other ratios? The axis label for panel c should not say “Whole rock” when glass is also plotted. Also, why no error bars for glasses?

Line 735. What bandwidth was used here for the kernel density plots?

Line 752. Please highlight bubble-bearing inclusions in all panels.

Line 766. Please identify inclusions with bubbles here (or note they are all bubble free). Please indicate uncertainties. Consider using GPa on the pressure axis for consistency, and add a depth axis (in km) in addition to pressure (and add this data to Figure 4). Finally, the legend refers to “PEC” but

you use “PEP” elsewhere in the manuscript.

Adam Kent

Referee #2 (Ken Rubin):

This is an interesting, timely, and rapid presentation of the results geochemical and petrological analyses of the still ongoing Fagradalsfjall eruption in SW Iceland. It is impressive that the authors have amassed so much data in so short of time, coordinating across many different researchers and Lab. I applaud the effort! And I find the results very interesting, as someone who specializes in just this sort of research.

Within the context of non-ideal writing and structure in the manuscript’s present form, the main results appear to be (1) sampling of magma that has experienced relatively little shallow differentiation, better preserving mantle signatures than most Icelandic eruptions and (2) relatively large geochemical variation in the opening phases of the eruption (although I disagree that these are the “largest ever observed”, as asserted repeated by the authors, presumably as part of the “hook” to interest Nature – this is an overstatement and displays a lack of familiarity with the literature, at the worst; note: I discuss several relevant papers in the detailed comments below, but these are just a subset that came to mind right away). A third result (3) relates to a possible (read: “somewhat speculative”) assertion for magma storage below the Moho for this eruption. Broadly speaking, while interesting to me as a specialist, the manuscript does not do an exceptionally good job of contextualizing this eruption for readers. For instance, the authors variably referring to the site as a Mid ocean Ridge, a Hotspot, and some uniquely Iceland hybrid where it convenient for them to make a particular point, but it leaves a manuscript without a solid geological underpinning or significant rationale for the study, and touchstone for comparison to other relevant eruptions. And this looseness permeates into the examples chosen for comparison to, which I personally don’t think are the most relevant, or help make the case that we are learning about processes that related to specific styles and locations of volcanism.

Additionally, as written, the manuscript suffers from several deficiencies that I believe could be easily amended in revision, but, when combined with the technical issues, preclude my strong endorsement for publication at the present time in the present format. Non-technical issues include a “jargon” approach to petrological (e.g., “primitive-evolved”, “high MgO”, etc.) and geochemical (e.g., “enriched” vs “depleted”) descriptors; a poorly/unevenly sampled literature (with many relevant papers missing and several ones of dubious relevance cited); perhaps the presenting of too much data of marginal relevance for a generalist reader; and unclear set up of the relevance of the study, for instance including but not limited to assertions about the tectonic setting and relevance of the eruption (as noted above),

Overall, I think this manuscript has the potential to be of interest to Nature readers, but probably not as written. The topic is potentially of interest to a broad audience such as Nature enjoys. Nature and Science together have published “similarish” papers on other notable recent eruptions (e.g., Kilauea 2018, Mayotte), so it obviously happens, occasionally. Yet, besides the main technical and

content topics, which I describe in more detail below, the manuscript is not written for a Nature audience. It is data rich and description poor, and not well referenced. I fully appreciate the challenges of writing for this venue, but if other referees and the Nature editorial staff agrees, I think the authors could rethink their approach here, writing a more compact and focused paper that drives home a few key points (notably, 1 and 2 from my first paragraph), and leaving the rest of the data and secondary interpretations to either be presented here in supplement and not discussed, or spread across other papers, or both.

Presently, the manuscript gives us a lot of data in rapid succession and with a huge number of confusing multi-panel images, which I don't think serves the authors well. I remember when Nature papers were just a few well-chosen figures with representative content, well described and contextualized for a generalist audience. What we seem to have here is content prepared for a full-length discussion a la J.Pet of G-Cubed, subsequently condensed to a few descriptive paragraphs plus supplements that will mostly be understandable only to specialists.

Additionally, several aspects of the data are interpreted at the bitter edge of realistic analytical uncertainty (e.g., Si content, Sr isotopes) combined with misrepresentation of data quality using 2SE internal statistics for uncertainty bars despite presenting accuracy and reproducibility analyses that exceed those ranges by 10x or so, which is right at the edge of variation in many of the presented parameters. I don't question the analytical capabilities or sensibilities of the authors or the study, but at face value, I think more effort should be devoted to discussing the data suitability and limitations relative to interpretations. I have likewise found myself in the awkward position of interpreting signals I think should be there, and probably are, but are not fully resolvable within realistic uncertainty bounds. I have found that some discussion of the limitations, alternate interpretations, analytical nuances, and scenario testing can be helpful in conveying a sense of why specific interpretations are being made, but it starts with a more realistic uncertainty assessment. For instance, have the authors considered what incorporating realistic uncertainties does to the stated ranges in geochemical and petrological parameters?

I challenge the authors to think a bit more strategically about how to get across their key points. For a revision for a short format generalist journal (Nature or otherwise), I strongly suggest subsampling the data for the most salient points and explaining them more fully, while resisting the urge to report on every geochemical tidbit analyzed. The C/S results, for example, provide weak constraint on depths, and yet get 2 longish paragraphs on the behavior of these gases in magmas, while the main petrologic data (majors and crystals) and geochemical source indicators (trace element and radiogenic isotope) get very brief mention, and no summary of systematics for the generalist. Perhaps the C/S result is suffering from too short a discussion, and might be better addressed in another paper?

Aloha, Ken Rubin

Beyond these general comments, I also made many notes on the manuscript itself as I read it, which I have transcribed here, perhaps with incomplete attention to grammar and spelling, given the accelerated review timeline. I apologize ahead of time if anything is unclear. I reference line manuscript numbers for these comments, presented here in numeric order, rather than relevance

order

Line 38-39: “the roles of centralized crustal magma reservoirs - in many, but not all cases. Plus, this assertion seems to set the tone that the paper is about Icelandic central volcanoes, not more MOR like settings. This would be a good place to describe the range of Icelandic and MOR magma supply and eruption conditions, properly referenced, and perhaps also to mention that some MORs (e.g., the MAR) exhibit hourglass segments some have interpreted as Icelandic style central volcanoes, whereas most of the intermediate, fast, and superfast MOR segments in the Pacific distinctly lack this feature (with Axial volcano being one exception)

Line 45 and throughout: “high-MgO”. Define. This, I am always struck by what counts as high MgO in Iceland, as compared to MORB

Line 48-49 and elsewhere: “composition changed at a rate unprecedented for basaltic eruptions globally.” I am dubious. Look at figure 2 of Gansecki et al., 2019 [Science 366, eaaz0147]. There we find 8 wt% variation (from 2 to 10 %) in MgO, all of it expressed within the first two weeks of eruption initiation. Even discounting differentiated compositions from fissure 17, the range is 4 to 10 wt %. In fact the ranges reported here in Iceland roughly equal those in typical, individual Pacific MOR eruptions on the EPRR and JdFR (see for example figure 1 of Rubin et al., 2001, Earth and Planetary Science Letters 188, 349-367

Line 51: enriched in what sense? Remember, Nature is a generalist journal.

Line 54-55: “source, providing new insights into the rates at which magmatic aggregation processes occur.” To my mind the observations are more like a confirmation of trends observed elsewhere, as referenced.

Lines 56-58: I am generally not keen on this opening paragraph. What point is trying to be made? Is the Reykjanes ridge being compared to a typical MOR? If so, state as much and qualify, as necessary. It appears as though this section is meant to say that direct observations at ridges are limited, so let's study and Iceland example. And yet the differences, especially in crustal thickness, and in magma production rate, complicate this. Plus at least 1/2 of the references cited are only loosely relevant, and I note that none of the literature that looks at active MOR eruptions from compositional change perspectives are cited, making the last sentence problematic.

Lines 64-68: Again, a problematic analogy, as both of the noted caldera/rift zone eruptions bear far more resemblance to ocean island volcanism than they do to MOR volcanism. Isn't this an apples and oranges comparison for ridges and non-ridge systems? How would a generalist, or even someone not steeped in Icelandic geology, appreciate the distinctions?

Line 78: “commonly referred to as volcanic systems”. Referred to as such in Iceland . Elaborate.

Line 101: “ primitive” This term is not meaningful to a general audience, and is petrologically dubious in a case of polybaric melt generation and differentiation. My suggestion is describe magmas as higher and lower MgO, and to explain what that means, what processes are encompassed, and how in some cases this might be seen as “evolution”.

Line 104: “evolved”: same comment, define and explain

Line 111: SiO₂: It is questionable as to where or not this is resolved outside of uncertainty

Line 112-114: “ Together, these changes in major element chemistry suggest a change toward greater depths (lower Na₂O/TiO₂) and lower degrees (higher K₂O/TiO₂) of melting over time.”: Is it worth a mention of why this is not a shallow differentiation signature? Also, it is risky to imply that this comes from an active deepening of the magma source. it could just as easily be a shift in proportional sampling of two discrete reservoirs, initially supplied with the two compositions simultaneously, followed by mixing thereof. Finally, how about a nod to Klein and Langmuir in the references?

Line 118-121: Isotopic “evolution” of the magmas: It is problematic that there are only 2 samples with all three radiogenic isotopic systems measured in them, and only 4 with both Sr and Nd. ALSO... the uncertainties reported in the supplement and shown on the images are unrealistically low, representing internal analytical standard errors, not true uncertainties. For instance, the reproducibility of 51 NBS 981 analyses reported as standard error, are much higher, and higher still, about 90 ppm, if unwrapped back to a standard deviation. The extremes of the Sr values are arguable indistinguishable given the standards data. Now, the apparent correlation with Nd isotopes in 4 samples is good, but this is by no means definitive. Especially in these sorts of generalist papers, it is important to emphasize what the state of the art, analytically speaking, is, and to honestly assess sources of uncertainty and reproducibility. I recall having to make similar arguments about barely resolvable Sr-Nd-Pb values at Kilauea about 20 years ago.

Line 122-123: “been observed at other oceanic hotspots where”. I thought the authors were writing this as if this was a MOR eruption site, for which the Hawaiian comparison is not good. In fact, even in Hawaii, the comparison to Puu Oo is a particularly poor one. As a minimum the authors should explain the setting, the comparisons, and the salient points they are attempting to convey. it is quite a different thing to talk about mixing of source signatures at hot spots and at MORs, with perhaps the most applicable examples coming from plume influences ridges, such as the Galapagos Spreading Center and the HUMP region of the EPR, where individual eruption deposits have been mapped and samples for within flow variations (e.g., see Colman, A. et al. (2016) Magmatic Processes at Variable Magma Supply Along the Galápagos Spreading Center: Constraints from Individual Eruptive Units, *Journal of Petrology*, 57(5), 981-1018, doi: 10.1093/petrology/egw032 and Bergmanis, E., et al. (2007) Recent Eruptive History and Magma Reservoir Dynamics on the Southern East Pacific Rise at 17.5°S, *Geochem. Geophys. Geosyst.*, 8, Q12O06, doi:10.1029/2007GC001742

Line 125-126: “However, the rapid compositional change in the case of the Fagradalsfjall eruption is unprecedented globally.”: I fundamentally disagree I don't think the authors are not at that familiar with the literature.

Line 130-135: I find this a very poor example for comparison, two primary reasons. a) the early phases of Puu oo have demonstrably been affected by crustal contamination with 1982 summit magma and b) it is unrealistic to compare a 35 year long, huge volume caldera driven rift zone

eruption, which developed a buffered, steady state magma supply condition for all of the 90s, 2000s and 2010s, limiting the amount of variation.. Far better examples would be shorter events, such as the aforementioned 2018 LERZ eruption, or the aforementioned SEPR and GSC MOR eruptions, or other petrologically well studied individual MOR eruptions such as 9 50N EPR (Goss et al., G-cubed, 2010) or the last two at Axial Volcano (Clague, D. et al., (2018) Chemical Variations in the 1998, 2011, and 2015 Lava Flows from Axial Seamount, Juan de Fuca Ridge: Cooling During Ascent, Lateral Transport, and Flow, *Geochemistry, Geophysics, Geosystems*, 19, 2915–2933. <https://doi.org/10.1029/2018GC007708>.)

Line 155-158: OHMI - Are we sure these unusual MI aren't in xenocrysts? Or, where might these melts sit and be picked up?

2 Paragraphs, lines 160-188: This longish section could be greatly reduced to a sentence or two (as discussed in the last sentence of the 2nd paragraph), given the looseness of the constraints. In other words, it doesn't add that much to the paper, and it consumes a lot of space that could instead be used to describe some of the other attributes more fully, especially exoneration of extant literature on within eruption deposit mantle and crustal geochemical signals in MORB.

Line 206-207: “ In the case of the latter, and in agreement with observations from global-scale variations in MORB” This could easily be elaborated upon, given the richness of the exiting MORB literature on the topic, and arguably, the limited applicability of observations at Holuhraun given the logical construct the authors themselves use, which is that Fagradalsfjall is a “MOR” like eruption where as Holuhraun was not MOR like. In fact, the arguments in ref 32 are based on a combination of global and individual eruption studies, one of the latter of which is described in detail, and which shows much greater within eruption compositional variability than the current example.

Line 215-222: It would be better if the ranges and rates of variation in time were defined and described, for both this eruption, and other notable Icelandic and MOR examples, as studied for instance, by MacLennan and others. For instance, how do ranges of MI compositions vary with MgO as compared to Borgarhraun or the high Mg mamgas at Peistareykir?

Line 224: “the entire spectrum...” This is the anticipated result in my opinion (as discussed for instance in ref 32), but nice to see it playing out in the data.

Line 237-239: I would be surprised if it weren't . But anyway, such a statement should be elaborated upon. How common or uncommon is it for collocated repeat eruptions from a restricted area to show completely different mantle source signatures?

Line 252-253: “Second, for the first time, rapid partial drainage of a sub-Moho magma reservoir has been monitored in near-real time. “ Note, the 2005-6 EPR eruption described in the aforementioned Goss et al. paper was also hypothesized to have had a component of magma storage in a submoho reservoir identified and described by Dunn et al., based on seismic tomography.

Author Rebuttals to Initial Comments:

We are very grateful for these expert critical reviews. Below we add our detailed response to the referees' comments, which we have strived to address diligently. The overall thrust has been to sharpen the rigour of analysis (appraisal of uncertainties) and develop the immediate and wider implications of our key findings.

Referee #1 (Adam Kent):

Review of "Rapid evolution of a deep magmatic system revealed by the Fagradalsfjall eruption, Iceland" by Halldórsson et al.

The manuscript reports a large data set obtained from geochemical, petrological and gas measurements from the recent Fagradalsfjall eruption on the Reykjanes Peninsula of Iceland. The paper presents data for this eruption that will be tremendously valuable for future workers in a variety of areas – including volcanology, mantle geochemistry, igneous petrology, and hazards. The work is also certainly novel as this is an exceptional eruption.

The manuscript makes the reasonable case that the eruption at Fagradalsfjall is different to other recent rift eruptions in Iceland and elsewhere, in that there is little evidence for shallow crustal storage, with magmas coming directly from the upper mantle or lower crust. This means that there is lesser crustal processing, and thus the opportunity to see mantle processes and mantle derived compositions with more fidelity. Geochemical variations also suggest that there was mixing between magmas derived from distinct enriched and depleted mantle domains and a rapid shift between these compositions during the ongoing eruption. Although the recognition of rapid transport of magmatic components from the mantle or lower crust is not completely new, and rapid transport of phenocrysts from the mantle or lower crust to the surface has been documented elsewhere in Iceland (see for example the work of Much et al. Nature Geosciences 2019), the opportunity to observe and fully document an eruption of this type is certainly a major opportunity – particularly for understanding the ultimate control that mantle processes might have on eruptions.

We thank Adam Kent for his positive and supportive assessment of this work.

My major comments on the manuscript are below, followed by some more minor issues.

Synthesis: I very much appreciate the uniqueness of this eruption, the size and quality of the data sets presented, and there is significant opportunity for learning something new, but I also had trouble seeing what were the major scientific insights that stem from these observations and data. As such this paper reads more to me as an exceptional documentation of this eruption, but less synthesis of what it tells us that is new (or confirms other existing ideas) about ridge systems. With some quibbles (see below) I broadly accept that the erupted magmas come from the upper mantle or lower crust, with little or no shallow crustal processing, but I do not feel like I learned something new about how mid ocean ridge magma systems work. Given the importance of the paradigm of the melt lens and shallow processing for many magmas in mid ocean ridges show I feel it is reasonable to expect a greater level of synthesis here.

We much appreciate this commentary. Since submitting the original manuscript (while the eruption was still ongoing!), we have had more time to fully assess the implications of our data. Accordingly, the manuscript now has clearer conclusions providing near real-time insights into how mid-ocean ridge magma systems work.

The importance of this eruption, and our data, is that it demonstrates more rapid changes in the composition of an erupting magma within in an Icelandic volcanic system than thought possible. The extreme mantle-derived compositional heterogeneity revealed in bulk rock chemistry was manifested at the surface surprisingly rapidly. We agree that previous studies of ancient eruptions have demonstrated (1) deep storage, (2) rapid transport from near-Moho to surface, and (3) extreme mantle-derived compositional heterogeneity. However, the eruption at Fagradalsfjall not only provides strong evidence for all of the above, but also demonstrates that geochemical proxies, which signify different mantle compositions and melting conditions, changed at a rate unprecedented for individual basaltic eruptions globally (in this case, magma mixing acted to increase (not decrease) the compositional diversity of erupted materials).

We anticipate our work will provoke future studies. For example, our observations suggest the following: (i) magmas can mix and erupt in a single event when extracted from deeply seated (near-Moho) reservoirs, (ii) mixing between two or more near-Moho magma reservoirs (lenses) is fast and on the order of a few hours/days. This has significant implications for physical / fluid-dynamical models of magmatic systems.

Finally, the fact that we can reconcile petrological constraints with gas measurements and reinforce the message of deep storage and rapid ascent is, to our best knowledge, unique for an eruption at a spreading ridge. To reflect this and to better highlight the major scientific insights, we have restructured and rewritten large parts of the manuscript.

See lines 39-58 (abstract) and lines 175 to 203 in the revised version and lines 235 to 274 as examples.

Interpretation as liquid: One important issue that needs clarification is the interpretation of the whole rock compositions as a liquid, which happens both explicitly and implicitly throughout the manuscript. This assumption is difficult to reconcile with the observation of abundant mafic and zoned phenocrysts, and many mafic magmas are mixtures of evolved liquids and antecrysts. I was surprised that there is no clear discussion of this important issue, and without that I have some trouble fully accepting some of the barometry and other key parts of the interpretation. One essential piece of information lacking in this respect is the modal proportions of crystals that are present and if these proportions vary through time. It is essential to provide these – and if possible, give the proportions of different crystal types identified. Variations in whole rock chemistry may well reflect variations in modal crystal proportions through time, rather than changes in some primitive igneous liquid.

We agree that our approach to interpret whole rock compositions as a liquid created some confusion. This assumption was made implicitly by application of the OPAM barometer to

the whole rock compositions. We have now removed these pressure estimates from the discussion and restrict the application of the OPAM barometer to clean glasses (i.e., glasses free of microphenocrysts) and melt inclusions (see below). In addition, we focus only on cpx-glass pairs where (i) for which textural evidence indicates equilibrium, (ii) and where pressure estimates confirm equilibrium tests from Neave and Putirka (2017) and Neave et al. (2019). The original conclusion stands: collectively they all point towards a near-Moho storage zone.

We accept that crystal modal abundance data would be valuable, and will be addressed in future work as part of a more comprehensive assessment of whole rock chemistry. However, we can use the data presented in this manuscript (both whole rock and glass data) to place first order constraints on the importance of this potential issue.

- Both MgO and TiO₂ contents of both glass and whole rock samples increased with time, which is inconsistent with increased modal proportions of olivine.
- In contrast, CaO and Al₂O₃ contents both decreased with time, which is inconsistent with increased modal proportions of plagioclase.
- Thus, the positive correlation between MgO and TiO₂ contents but negative correlation between CaO and Al₂O₃ and TiO₂, evident in our whole rock data, indicate that an increase in modal crystal proportions, such as those of olivine and plagioclase—which are both present as macrocrysts (<5%, observed visually) in the lavas—does not control the major and minor element chemistry.

In addition, and perhaps most importantly for this manuscript, we note that the geochemical proxies used to assess source shifts (K₂O/TiO₂, La/Yb and radiogenic isotopes) are highly insensitive to crystal removal or additions during crystal fractionation from high MgO liquids, as observed in the Fagradalsfjall eruption. This includes PEP corrections applied for raw melt inclusions data. The dramatic temporal changes recorded in these proxies cannot be explained by varying modal proportions of crystals. Additionally, where possible, glass analyses are used rather than whole rock. In the revised version, we more clearly highlight that variations seen in the chemistry over time are first and foremost noteworthy because they are observed in geochemical proxies for mantle composition and melting process—such as K₂O/TiO₂, La/Yb, ²⁰⁶Pb/²⁰⁴Pb—which are immune to shallow level processes, including variations resulting from varying modal proportions of crystals.

The bottom line is that our primary conclusions are not sensitive to crystal loads, and conclusions dealing with barometry are now based on glasses and melt inclusions.

See lines 39-58 (abstract), lines 115 to 124 and lines 159-173.

Barometry: The barometry is an important part of the interpretation that magmas do not experience shallow crustal residence, however there are some limitations with respect to the data supplied that prevent a full assessment of this interpretation, mostly revolving around assessing uncertainties. There is increasing awareness that uncertainties in calculated pressures from barometry can be much larger than the canonical uncertainties used here (such as the 1.4 kbar uncertainties for cpx-liquid from the original work of Putirka - e.g. see Wieser et al. 2021 – GSA Meeting abstract) as these do not fully take into account

propagation of measurement uncertainties, differences in electron probe calibrations between laboratories, as well as the uncertainty derived from the calibration itself. Although it is perhaps unfair to request that these authors hold themselves to a higher standard than most other currently published work, at the very least they should provide enough metadata that these uncertainties can be independently evaluated, and there is a general level of incompleteness and inconsistency with the supplied metadata and uncertainties (see below). For example the calculated pressures for melt inclusions should also include uncertainties related to calibration and measurements (or be specific about exactly what these uncertainties represent).

We recognize the importance of rigorous assessment of the uncertainties in the barometry. We further agree with the reviewer that such a comprehensive assessment of barometric uncertainties is beyond the scope of this article (and we keenly await publication of the results by Wieser et al.), but we have provided as much meta-data as possible in the supplementary data and tables and have revisited our treatment of uncertainties.

We refer the reviewer to our response to the “Metafile” comment below, where we demonstrate that all data (including key aspects of relevant analytical uncertainties) are now reported in full. It is therefore, our hope that this clear presentation of all data relevant for our barometric models will allow readers to make comparisons with their own datasets and use their own models for our data if so desired.

See revised barometric section (lines 159-173) and a summer method section on our barometric models (line 587-672). See also our response to the “Metafile” comment below.

I am less familiar with the OPAM barometer but I also wonder what measurement uncertainties do when propagated through this calculation. In addition, all code used to determine pressures should also be released (I believe this is nature policy). I am not aware that the OPAM code is publicly available.

The OPAM code, written by John Maclennan, was introduced in a prior publication involving several co-authors of the present manuscript (Hartley et al., 2018), where the following sentence can be found “*An example Python script using our implementation of the OPAM barometer is available from the authors on request.*” However, a Jupyter Notebook is now supplied in Supplementary Information to this manuscript with the scripts used to perform the calculations. It includes the script and input files needed to reproduce the calculation.

As described in Hartley et al., 2018, proper error assessment of the uncertainties is built into this model. In the case of our data, all errors are reported in Supplementary data table 9. This estimate does take into account the analytical error (see Hartley et al., 2018). Note also, that in this same table, we specify the number of samples that pass the probability filter of the code. Only such samples are reported and discussed in the text. See table footnotes for details.

See revised method section summarizing our OPAM calculations (line 587-607)

I am also confused why the pressures of melt inclusions are not included in Figure 4 as they should also be useful for this interpretation.

Melt inclusions OPAM pressures were included in Figure 4a, and we have tried to highlight them more clearly. The minimum constraint from CO₂-H₂O saturation pressures of melt inclusions is also shown now. See also further discussions below on these same calculations – see comment regarding line 766 in the original submission.

Finally, I also note inconsistency in the units used for pressure – where between text and figures kbar, bar and GPa are all used (I recommend using GPa as it is an SI unit).

We agree, and we apologize for this inconsistency in pressure units. All pressures are now expressed in GPa.

Metadata: As noted above there is a general incompleteness and inconsistency in the metadata supplied to support the various data sets. This includes details on calibration standards, count times and secondary standards. I recommend including the following:

1. A list of actual calibration standards used for each element and count times for every element in all electron probe measurements (these are noted as being in Table 1-5 in the text but I could not find them).
2. Reported uncertainties for every element in every electron probe analysis (as provided for LA-ICP-MS and SIMS analyses for example).
3. Complete analyses of all materials analyzed as secondary standards for all analyses. At present there are no such data that I can see.
4. A clear indication how uncertainties provided for SIMS and LA-ICP-MS were determined and a clearly labelled and consistent use of 1s or 2s for ALL uncertainties in text and figures.
5. There is also very inconsistent use of error bars to reflect uncertainties in figures. Some figures have errors but not for all data, some have no error bars at all. Please include a representation of uncertainties in all figures where data is plotted, either as individual error bars or representative error bars.

We fully accept the points made concerning standards and uncertainties and have addressed them with the following changes:

1. We now include a list of calibration standards and count times for every element in all electron probe measurements in the relevant Supplementary Data Tables 5 (pyroxene), 6 (olivine), 7 (plagioclase), 8 (spinel) and 9 (glass). See also revised EPMA methods section where we guide readers to the correct tables (see Supplementary Tables 5 to 9).
2. We now report uncertainties for every element in every electron probe analysis in these same tables.
3. We also include analyses of all secondary standards in these tables.
4. Trace element and major volatile element analyses of melt inclusions and a few glass grains were done with SIMS only and at the NordSIMS lab. We have also revised the EPMA methods section.
5. Error bars have been added to all relevant figures. Note that we include only the errors on the new data presented here, but not literature data that we use for comparison.

We also apologize for not including this critical information in the original submission and thank the reviewer for pointing this out.

See also a revised Methods section, including additional information provided in the Supplementary Information.

Other comments linked to line numbers:

Line 113. It looks like K_2O/TiO_2 correlates broadly with isotopic composition – if so, how is sensitive to degree of melting? I suspect it is difficult to deconvolve degree of melting and source variations here.

The reviewer is correct that this observation suggests that K_2O/TiO_2 is also sensitive to degree of melting. In fact, and as also indicated by the reviewer, a major challenge in igneous geochemistry is to disentangle the effects of source and process on the ratios of incompatible elements, such as K_2O/TiO_2 . This has in fact been addressed in several recent on Icelandic basalts so our observation is not a new one (good examples being recent papers by Stracke and collaborators, Peate et al. (2009/2010) and Shorttle and Maclennan (2011)). However, we note that radiogenic isotope ratios do not change during partial melting and reflect real isotopic variation in the mantle. Contemporary models of mantle melting show that different lithologies enter the melt as a function of the degree of partial melting. More fusible, enriched lithologies enter first, followed by more refractory, depleted lithologies. This generates correlation between the degree of partial melting, incompatible element ratios (sensitive to both source lithology and degree of partial melting), and radiogenic isotope ratios. As a result, such correlations between indices of partial melting and source are widely observed throughout Iceland and in basalts globally. This idea is represented in Figure 4b, showing deep melts form at low degrees of partial melting and are enriched, and shallow melts form at high degrees of partial melting and are depleted.

The bottom line is that there is rather dramatic (on the scale of the Reykjanes Peninsula) radiogenic isotopic variability in the Fagradalsfjall eruption, which highlights mantle source variability. Thus, while incompatible element ratios like K_2O/TiO_2 are sensitive to the melting process, radiogenic isotope variability in the eruption tells us that the mantle source is heterogeneous. It is therefore important to use K_2O/TiO_2 and radiogenic isotopes together, but this is of course not always possible as we can't measure radiogenic isotopes on as much material as we can K_2O/TiO_2 .

See lines 39-58 (abstract) and a new Figure 4, including captions.

Line 148 and elsewhere. It would be good to understand the uncertainties associated with the corrections for post entrapment crystallization (PEC) and also the sensitivity of elements to this correction. For example, MgO in olivine hosted melt inclusions is very sensitive to PEC (Kent 2008 RiMG vol 69), and even small variations in olivine addition produce large MgO variations. Using corrected compositions for further calculations – such as thermobarometry – could also be associated with large uncertainties in P and T, so this issue should be discussed.

We agree that understanding potential uncertainties associated with PEP is very important for any melt inclusion data. In light of this we decided to simplify discussions involving our melt inclusions data considerably. When comparing Fagradalsfjall whole rock and glass compositions to the melt inclusions from the same eruption, we rely only on ratios of two incompatible minor and trace elements, such as K_2O/TiO_2 and La/Yb (Figure 3), because these are not affected by crystallization and therefore, PEP corrections. This is very important as this approach allows us to interpret our melt inclusion data without having to carefully assess uncertainties associated with the PEP corrections. MgO contents of these same inclusions are however plotted only to provide an assessment of the degree of evolution (Extended Data Figure 2c-d) but because of the approach describe above, we consider comprehensive assessment of uncertainties associated with the PEP corrections beyond the scope of this article

We also agree that melt inclusion-derived OPAM pressures must have a larger uncertainty than those derived from matrix glasses due to uncertainties in post-entrapment corrections (see response above regarding assessment of the uncertainties in the barometry, including OPAM errors). Accordingly, we have revised the manuscript to first and foremost emphasize the P-T results derived from matrix glasses. Nevertheless, we have preserved the P-T results from melt inclusions in the manuscript and in figure 4, as we agree with the reviewer that melt inclusions remain useful for this interpretation (see reviewer's comment above). Indeed, pressures and temperatures derived from PEP-corrected melt inclusions with MgO content similar to matrix glasses overlap (Fig 4a and Extended Data Figure 10). See also our responds to errors associated with the OPAM model we use.

Finally, we refer to a fully updated Supplementary Data Tables where all data are now reported in full. This includes key aspects of relevant analytical uncertainties as well as details regarding the PEP procedures adopted (See Tables 10 and 11). We hope that reporting these details will allow readers to make comparisons with their own datasets and/or perform detailed assessment of the uncertainties of this dataset if so desired. In addition, and using data provided in the Supplementary Data Tables, we now include representative analytical error bars on all figures.

Line 162. This is a good place to note that saturation pressures only reflect bubble free melt inclusions, as this is important.

Pressures are calculated for inclusions both with and without bubbles but as suggested by the reviewers, we now identify bubble-bearing inclusions specifically on all relevant plots (i.e., Extended Figures 7 and 8). However, regardless of whether a vapor bubble is present, the CO_2-H_2O saturation pressures still only provide a minimum constraint on saturation pressure due to decrepitation (Maclennan, 2017; Matthews et al., 2021).

Line 221. Although the range of compositions observed is large, it is also important to note that you do not get to equivalently sample all stages of earlier Reykjanes Peninsula eruptions, and more variable compositions could be obscured by being covered by later erupted material. What range of samples would you get if you just were able sample material present only at the later stages of the eruption?

This is interesting topic but we would like to point out that in Iceland's neovolcanic zones, the older subglacially erupted units have often been eroded such that the hyaloclastite mount is dissected, exposing the initial stages of eruption. This means that we can, in many places, sample material from close to the beginning of events. In the case of the Fagradalsfjall eruption, it remains challenging to address what material we will have access to in the future because it is not yet clear that this episode is over. Recent earthquake swarms and geophysical evidence for renewed deep magma reservoir growth in the region (as of December 2021) suggest the possibility of further eruptions.

However, lavas from quite early in the eruption (Eruption day 17/ April 5th) while still exposed on the surface, now represent a very small fraction of the exposed lava. Very detailed sampling of the lava field would need to be performed to find these early, geochemically more depleted flows.

Line 538. This additional correction sounds like it would introduce further uncertainty – has this been included in the reported uncertainties?

Good point. The uncertainty associated with these measurements have been revised following this suggestion by the SIMS experts of our team (Martin J. Whitehouse and Heejin Jeon). Considering this additional correction resulted in very minor increase in analytical uncertainties (few % only). See a revised SIMS methods summary in main Methods but a detailed description in the Supplementary Information. In lines 541-545 we explain the following that relates to this issue:

“Uncertainties were calculated from the signal variability of the sample analysis, the scatter along the calibration curve, and the uncertainty of the composition of the calibrating standards. Overall analytical uncertainties for H₂O are less than 4%, and less than 8% for CO₂, based on repeatability of the calibration reference materials.”

Line 596. Note that TiO₂ will be impacted by post entrapment crystallization in that it will be increased in concentration (as will all incompatible elements) by crystallization.

We agree. As pointed out in our response above, in this revised version we emphasize that when comparing Fagradalsfjall whole rock and glass compositions to the melt inclusions from the same eruption, we rely only on ratios of two incompatible minor and trace elements, such as K₂O/TiO₂ and La/Yb, because these are not affected by crystallization. This is important in light of the uncertainty associated with correcting for post-entrapment processes. Therefore, this is no longer relevant as we do not draw any conclusions from incompatible element contents in the manuscript.

Line 629. Please provide a source for these quoted uncertainties and state which uncertainty components that they include.

We now cite the source for the thermometer used (Putirka, 2008), which is relevant for the discussion of the uncertainty associated with the calculations.

Line 706. Why are glass analyses only shown for K₂O/TiO₂ but not the other ratios? The axis

label for panel c should not say “Whole rock” when glass is also plotted. Also, why no error bars for glasses?

Good point. We do not plot any other minor element ratios than K_2O/TiO_2 and when relevant, we specify if we are plotting whole rock or glass compositions, or a combination of the two.

Line 735. What bandwidth was used here for the kernel density plots?

The bandwidth of 0.2 is now specified in the figure caption.

Line 752. Please highlight bubble-bearing inclusions in all panels.

We now highlight bubble-bearing inclusions in the two panels reporting CO_2 data. The presence of bubbles appears not to be important for S and H_2O , only CO_2 , and therefore we highlight bubble-bearing inclusions on plots involving CO_2 .

Line 766. Please identify inclusions with bubbles here (or note they are all bubble free). Please indicate uncertainties. Consider using GPa on the pressure axis for consistency, and add a depth axis (in km) in addition to pressure (and add this data to Figure 4). Finally, the legend refers to “PEC” but you use “PEP” elsewhere in the manuscript.

Agreed. We have revised this figure (Extended Data Figure 8) following these suggestion. The uncertainty propagated to saturation pressure from the analytical uncertainty in CO_2 is negligible (<10% for the majority of inclusions) compared to model calibration uncertainty. As described by Wieser et al. (accepted manuscript), this calibration uncertainty is largely unquantifiable, and is best assessed qualitatively by looking at the discrepancy between models, as we show in Extended Fig. 8. In the Methods, we provide calculations that demonstrate the analytical uncertainty propagation (see lines 639-645). Including this uncertainty on the figures or in the data table would be misleading. In the main text, these data are used only in Figure 4, where we use the highest saturation pressure recorded as a minimum constraint on entrapment pressure. As a conservative representation of the uncertainty, we show the lower bounds on this, calculated using the Shishkina et al. (2014) model. In any case, the saturation pressures provide only a minimum bound on the entrapment pressure, due to crystal decrepitation (Maclennan, 2017; Matthews et al., 2021). A Jupyter notebook used to perform the calculations is now provided in Supplementary Information. Again, it includes the full script and input files needed reproduce the calculation.

See added text in the Methods, lines 627 to 645.

Adam Kent

We thank Adam Kent for constructive comments that have helped to greatly improve this manuscript.

Referee #2 (Ken Rubin):

This is an interesting, timely, and rapid presentation of the results geochemical and petrological analyses of the still ongoing Fagradalsfjall eruption in SW Iceland. It is impressive that the authors have amassed so much data in so short of time, coordinating across many different researchers and Lab. I applaud the effort! And I find the results very interesting, as someone who specializes in just this sort of research.

We thank Ken Rubin for his positive and supportive assessment of this work.

Within the context of non-ideal writing and structure in the manuscript's present form, the main results appear to be (1) sampling of magma that has experienced relatively little shallow differentiation, better preserving mantle signatures than most Icelandic eruptions and (2) relatively large geochemical variation in the opening phases of the eruption (although I disagree that these are the "largest ever observed", as asserted repeated by the authors, presumably as part of the "hook" to interest Nature – this is an overstatement and displays a lack of familiarity with the literature, at the worst; note: I discuss several relevant papers in the detailed comments below, but these are just a subset that came to mind right away). A third result (3) relates to a possible (read: "somewhat speculative") assertion for magma storage below the Moho for this eruption.

We thank the reviewer for this comment, for providing suggestions on how to better focus the manuscript, and for pointing out other studies to make comparisons with this dataset. We have taken his suggestions to heart in the revision process, which we hope the reviewer will note. We believe that we have a much better focused manuscript as a result and we have further justified and clarified the "hook" in the comments below. In this regard, we are indebted to the reviewer's comments which helped to significantly strengthen and focus our manuscript.

Our main focus is now on reviewer points (1) and (2), and we highlight that the context provided by barometry of near-Moho storage and mixing (point 3) is crucial for understanding their relevance to other magmatic systems, and we hope that the reviewer agrees that we have clarified these observations and justified our interpretations.

Broadly speaking, while interesting to me as a specialist, the manuscript does not do an exceptionally good job of contextualizing this eruption for readers. For instance, the authors variably referring to the site as a Mid ocean Ridge, a Hotspot, and some uniquely Iceland hybrid where it convenient for them to make a particular point, but it leaves a manuscript without a solid geological underpinning or significant rationale for the study, and touchstone for comparison to other relevant eruptions. And this looseness permeates into the examples chosen for comparison to, which I personally don't think are the most relevant, or help make the case that we are learning about processes that related to specific styles and locations of volcanism.

Thank you for allowing us to clarify this further. We agree that referring to Iceland as a Mid Ocean Ridge, a Hotspot, and some uniquely Iceland hybrid in our original submission

distracted from the main focus points. We are now more consistent about how we refer to Iceland as a unique locality as it allows us to explore a ridge in a subaerial environment.

Additionally, as written, the manuscript suffers from several deficiencies that I believe could be easily amended in revision, but, when combined with the technical issues, preclude my strong endorsement for publication at the present time in the present format. Non-technical issues include a “jargon” approach to petrological (e.g., “primitive-evolved”, “high MgO”, etc.) and geochemical (e.g., “enriched” vs “depleted”) descriptors; a poorly/unevenly sampled literature (with many relevant papers missing and several ones of dubious relevance cited); perhaps the presenting of too much data of marginal relevance for a generalist reader; and unclear set up of the relevance of the study, for instance including but not limited to assertions about the tectonic setting and relevance of the eruption (as noted above),

Thank you for allowing us to clarify this further. We have put much effort into improving the technical issues and have confidence they have been dealt with in a satisfactory and robust manner; see comment replies below as well as to Reviewer 1 (R1).

Frequently used petrological and geochemical terms have been defined in the revised manuscript for the benefit of a general scientific audience.

For example, in lines 101 to 104 we clarify the meaning of term “primitive” used very frequently in the manuscript.

*“Whole-rock MgO and TiO₂ contents range from 8.8–10.0 wt% and 0.95–1.12 wt%, respectively (**Extended Data Figure 2a**), where the high MgO content suggests the magmas were less processed in magmatic plumbing systems, and here referred to as “primitive”.”*

And in lines 130 to 135—which is the first time we introduce the term “enriched”—we define the meaning of the term:

*“The Fagradalsfjall lava also records a simultaneous shift towards more radiogenic Sr and Pb, and less radiogenic Nd isotope ratios, confirming that the deeply derived, lower-degree melts from later in the eruption sample a higher proportion of an enriched mantle source with higher incompatible trace element concentrations, and radiogenic isotope signatures indicative of long-term incompatible trace element enrichment (**Extended Data Figure 4**).”*

Overall, I think this manuscript has the potential to be of interest to Nature readers, but probably not as written. The topic is potentially of interest to a broad audience such as Nature enjoys. Nature and Science together have published “similarish” papers on other notable recent eruptions (e.g., Kilauea 2018, Mayotte), so it obviously happens, occasionally. Yet, besides the main technical and content topics, which I describe in more detail below, the manuscript is not written for a Nature audience. It is data rich and description poor, and not well referenced. I fully appreciate the challenges of writing for this venue, but if other referees and the Nature editorial staff agrees, I think the authors could rethink their approach here, writing a more compact and focused paper that drives home a few key points (notably, 1 and 2 from my first paragraph), and leaving the rest of the data

and secondary interpretations to either be presented here in supplement and not discussed, or spread across other papers, or both.

We have taken these suggestions to heart in the revision process, which we hope the reviewer will note. It is our view that the revised manuscript now goes significantly further beyond documentation of an exceptional eruption, offering new insights into how mid-ocean ridge magma systems work. For this, we followed the reviewer's suggestion and our main focus is now on his points (1) and (2). To place these observations into context, we have retained the presentation of data supporting near-Moho magma storage and mixing (point 3) and have provided a more clarified, succinct, and robust discussion around this. In particular, secondary observations and interpretations related to point (3) are now largely presented in the Method section. We believe that this has significantly improved the manuscript and again, we thank the reviewer for this guidance. If permitted, see also our respond to R1.

Presently, the manuscript gives us a lot of data in rapid succession and with a huge number of confusing multi-panel images, which I don't think serves the authors well. I remember when Nature papers were just a few well-chosen figures with representative content, well described and contextualized for a generalist audience. What we seem to have here is content prepared for a full-length discussion a la J.Pet of G-Cubed, subsequently condensed to a few descriptive paragraphs plus supplements that will mostly be understandable only to specialists.

The revised manuscript is more focused, including a more selective inclusion of figures.

See revised Figures 2 and 3 as examples of what we regard to be few well-chosen figures that are of the most importance for our data and synthesis.

Additionally, several aspects of the data are interpreted at the bitter edge of realistic analytical uncertainty (e.g., Si content, Sr isotopes) combined with misrepresentation of data quality using 2SE internal statistics for uncertainty bars despite presenting accuracy and reproducibility analyses that exceed those ranges by 10x or so, which is right at the edge of variation in many of the presented parameters. I don't question the analytical capabilities or sensibilities of the authors or the study, but at face value, I think more effort should be devoted to discussing the data suitability and limitations relative to interpretations. I have likewise found myself in the awkward position of interpreting signals I think should be there, and probably are, but are not fully resolvable within realistic uncertainty bounds. I have found that some discussion of the limitations, alternate interpretations, analytical nuances, and scenario testing can be helpful in conveying a sense of why specific interpretations are being made, but it starts with a more realistic uncertainty assessment. For instance, have the authors considered what incorporating realistic uncertainties does to the stated ranges in geochemical and petrological parameters?

We agree that this is an important point, raised also by the other reviewer. In the revised manuscript, we now include all of our secondary standard analyses and our assessment of the external uncertainty in the Supplementary Data file (Tables S1-S11) and other Supplementary information containing additional analytical details beyond those provided

in Methods. We also ensure the correct uncertainties are shown in all figures. Accordingly, we have made the following changes:

1. Significant improvement and expansion of all supplementary tables that includes all relevant analytical errors, standard data, and analytical details where necessary.
2. Errors bars have been added to data points in all relevant figures based on the propagated uncertainty of accuracy and precision of replicate analyses of secondary standard reference materials. We always specify if we are presenting 1 or 2 sigma error bars.

Having dealt with the technical issues of this comment, and in particular having added realistic error bars to all relevant figures, we hope that the reviewer agrees that the variation we observe is real. We also note that many of our analyses (e.g., trace element concentrations) were obtained in single sessions, meaning that the internal-reproducibility is the key parameter for robustly identifying heterogeneity within our dataset.

I challenge the authors to think a bit more strategically about how to get across their key points. For a revision for a short format generalist journal (Nature or otherwise), I strongly suggest subsampling the data for the most salient points and explaining them more fully, while resisting the urge to report on every geochemical tidbit analyzed. The C/S results, for example, provide weak constraint on depths, and yet get 2 longish paragraphs on the behavior of these gases in magmas, while the main petrologic data (majors and crystals) and geochemical source indicators (trace element and radiogenic isotope) get very brief mention, and no summary of systematics for the generalist. Perhaps the C/S result is suffering from too short a discussion, and might be better addressed in another paper?

Aloha, Ken Rubin

We thank Ken Rubin for his detailed and constructive comments that have helped us to improve the presentation of this data considerably and streamline the main messages.

Following his suggestions, we have streamlined the manuscript to focus on the key observations that reveal the unique insight Fagradalsfjall affords us of near-Moho mantle processes. Essential to providing the context with which to view these observations and their relevance to other magmatic systems is the barometric constraints on where the magma is derived and mixing is occurring. In recent years it has become ever clearer how difficult it is to obtain reliable barometric constraints for many eruptions (e.g., the ongoing work by Maclennan and Wieser); however, we show that multiple barometric techniques (melt compositions, gas compositions, cpx-liquid equilibria and melt inclusion saturation pressures) all point towards near-Moho magma storage and mixing. We provide full documentation of these constraints in the methods text, and have significantly focused their discussion in the main text, in accordance with the reviewer's suggestions (the C/S discussion is now two sentences rather than two paragraphs). Moreover, the fact that we can reconcile petrological constraints with gas measurements and reinforce the message of deep storage and rapid ascent is, to our knowledge, also something of a first for an eruption at a spreading ridge. Therefore, the barometric constraints provide not only an essential

context in which to place the other observations, but are also novel and insightful observations in their own right.

Beyond these general comments, I also made many notes on the manuscript itself as I read it, which I have transcribed here, perhaps with incomplete attention to grammar and spelling, given the accelerated review timeline. I apologize ahead of time if anything is unclear. I reference line manuscript numbers for these comments, presented here in numeric order, rather than relevance order

Thank you!

Line 38-39: “the roles of centralized crustal magma reservoirs – in many, but not all cases. Plus, this assertion seems to set the tone that the paper is about Icelandic central volcanoes, not more MOR like settings. This would be a good place to describe the range of Icelandic and MOR magma supply and eruption conditions, properly referenced, and perhaps also to mention that some MORs (e.g., the MAR) exhibit hourglass segments some have interpreted as Icelandic style central volcanoes, whereas most of the intermediate, fast, and superfast MOR segments in the Pacific distinctly lack this feature (with Axial volcano being one exception)

Agreed; this sentence as well as the entire abstract has been revised. See lines 39 to 58

We agree that more explicit contextualization of MORs, notably the recent activity at Axial seamount, makes sense. In the revised text we now point out that these observations are perhaps better described as being restricted to some closely observed recent Icelandic rifting events and that such processes have also been evidenced recently along some mid-ocean ridge sites. We refer the reviewer to Supplementary Figure 12 and 13, in which we summarize a comparison between the Fagradalsfjall eruption and other relevant eruptions.

Line 45 and throughout: “high-MgO”. Define. This, I am always struck by what counts as high MgO in Iceland, as compared to MORB

We agree with the reviewer that a better context for what we mean by high-MgO is needed here. We do, however, find this term unnecessary at this point and have decided to omit it from the early portion of the manuscript. However, at its first introduction, we hope to have provided a better context.

See lines 101 to 104

Line 48-49 and elsewhere: “composition changed at a rate unprecedented for basaltic eruptions globally.” I am dubious. Look at figure 2 of Gansecki et al., 2019 [Science 366, eaaz0147]. There we find 8 wt% variation (from 2 to 10 %) in MgO, all of it expressed within the first two weeks of eruption initiation. Even discounting differentiated compositions from fissure 17, the range is 4 to 10 wt %. In fact the ranges reported here in Iceland roughly equal those in typical, individual Pacific MOR eruptions on the EPRR and JdFR (see for example figure 1 of Rubin et al., 2001, Earth and Planetary Science Letters 188, 349-367

We fully accept this point. We are simply discussing trace elements and isotopes at this juncture, namely, proxies for mantle source heterogeneity, and we have modified the sentence accordingly. To ensure clarity, we have modified the manuscript title to better emphasize our observation of rapid switching in mantle-derived compositions and mantle source signatures over the course of the eruption.

See lines 47 to 49.

Line 51: enriched in what sense? Remember, Nature is a generalist journal.

Agreed. We now clarify what we mean by enriched. See also our response above.

See lines 130 to 135

Line 54-55: "source, providing new insights into the rates at which magmatic aggregation processes occur." To my mind the observations are more like a confirmation of trends observed elsewhere, as referenced.

This last part of the abstract has been entirely revised and it is our hope that it reflects reviewer's suggestions.

See modified text in lines 39 to 58 (abstract)

Lines 56-58: I am generally not keen on this opening paragraph. What point is trying to be made? Is the Reykjanes ridge being compared to a typical MOR? If so, state as much and qualify, as necessary. It appears as though this section is meant to say that direct observations at ridges are limited, so let's study an Iceland example. And yet the differences, especially in crustal thickness, and in magma production rate, complicate this. Plus at least 1/2 of the references cited are only loosely relevant, and I note that none of the literature that looks at active MOR eruptions from compositional change perspectives are cited, making the last sentence problematic.

Agreed. As discussed above, referring to Iceland as a Mid Ocean Ridge, a Hotspot, and some uniquely Iceland hybrid in our original submission distracted from the main focus points. We are now more consistent in referring to Iceland as a unique locality which allows us to observe mid-ocean ridge processes in a subaerial environment.

We have revised the opening sentence and it now reads:

While the Icelandic crust is thicker than typical oceanic crust¹, the subaerial exposure of the MOR permits continuous, real-time sampling of eruptions yielding critical insights into magma processes and timescales representative of their submarine counterparts.

See lines 69 and 71.

Additionally, we have followed the reviewer's suggestions and now cite more pertinent references. We have removed:

- Carbotte, S. M., Smith, D. K., Cannat, M. & Klein, E. M. Tectonic and magmatic segmentation of the Global Ocean Ridge System: A synthesis of observations. *Geological Society Special Publication* **420**, 249–295 (2016).
- Cannat, M., Cann, J. & MacLennan, J. Some hard rock constraints on the supply of heat to mid-ocean ridges. *Mid-Ocean Ridges: Hydrothermal Interactions Between the Lithosphere and Oceans* **148**, 111–149 (2004).
- Wanless, V. D. & Behn, M. D. Spreading rate-dependent variations in crystallization along the global mid-ocean ridge system. *Geochem. Geophys., Geosyst.* **18**, 3016–3033 (2017).

and added:

- Colman, A. et al. (2016) Magmatic Processes at Variable Magma Supply Along the Galápagos Spreading Center: Constraints from Individual Eruptive Units, *Journal of Petrology*, 57(5), 981-1018, doi: 10.1093/petrology/egw032
- Bergmanis, E., et al. (2007) Recent Eruptive History and Magma Reservoir Dynamics on the Southern East Pacific Rise at 17.5°S, *Geochem. Geophys. Geosyst.*, 8, Q12006, doi:10.1029/2007GC001742
- Goss et al., *Geochemistry of lavas from the 2005–2006 eruption at the East Pacific Rise, 9°46'N–9°56'N: Implications for ridge crest plumbing and decadal changes in magma chamber compositions* G-cubed, 2010
- Clague, D. et al., (2018) Chemical Variations in the 1998, 2011, and 2015 Lava Flows from Axial Seamount, Juan de Fuca Ridge: Cooling During Ascent, Lateral Transport, and Flow, *Geochemistry, Geophysics, Geosystems*, 19, 2915–2933. <https://doi.org/10.1029/2018GC007708>.)

This is also discussed below.

Lines 64-68: Again, a problematic analogy, as both of the noted caldera/rift zone eruptions bear far more resemblance to ocean island volcanism than they do to MOR volcanism. Isn't this an apples and oranges comparison for ridges and non-ridge systems? How would a generalist, or even someone not steeped in Icelandic geology, appreciate the distinctions?

We beg to differ on this point. In our view, these eruptions probably provide a better analogy to the lower magma flux MOR segments with central volcanoes than they do to OIB settings. We also point out that the Krafla Fires have played an important role in the understanding of MOR volcanic systems. We also refer to the review of Wright et al. (2012) who describe magmatic segments in Afar and volcanic systems in Iceland as spreading centers that are “*analogous to the second-order, non-transform offset segments observed on slow-spreading mid-ocean ridge*” as one way of justifying this analogy.

Line 78: “commonly referred to as volcanic systems”. Referred to as such in Iceland . Elaborate.

Agreed. In order to avoid confusion, we now focus on these systems being defined as intra-transform spreading centers, which better fits their overall characteristics and provides a better analogy to mid-ocean ridges (e.g., Fornari et al. 1989; Structure and topography of the Siqueiros transform fault system: Evidence for the development of intra-transform spreading centers, *Marine Geophysical Research* 11(4):263-299). Accordingly, we omit the term volcanic systems.

Line 101: “primitive” This term is not meaningful to a general audience, and is petrologically dubious in a case of polybaric melt generation and differentiation. My suggestion is describe magmas as higher and lower MgO, and to explain what that means, what processes are encompassed, and how in some cases this might be seen as “evolution”.

We agree and have made the relevant change throughout the manuscript. See also our previous response to a similar comment.

Line 104: “evolved”: same comment, define and explain

We agree and have made the relevant change throughout the manuscript. See also our previous response to a similar comment.

Line 111: SiO₂: It is questionable as to where or not this is resolved outside of uncertainty

Following the reviewer’s suggestion, we have chosen to move omit this figure as it does not deliver our most important message. We now focus on fewer, carefully chosen panels for clarity. See also our previous response to a similar comment.

Aside from the relevance of this figure, we point out that we have added a careful assessment of analytical uncertainty in the supplementary materials (see Supplementary Tables). From repeated measurements of the secondary standards, we can assign a robust uncertainty (external error) to all relevant figures. We provide all the necessary information for readers to make comparisons with their own datasets if so desired. In addition, and using data provided in the Supplementary Tables/Information, we now include representative error bars on all figures.

Line 112-114: “ Together, these changes in major element chemistry suggest a change toward greater depths (lower Na₂O/TiO₂) and lower degrees (higher K₂O/TiO₂) of melting over time.”: Is it worth a mention of why this is not a shallow differentiation signature? Also, it is risky to imply that this comes from an active deepening of the magma source. it could just as easily be a shift in proportional sampling of two discrete reservoirs, initially supplied with the two compositions simultaneously, followed by mixing thereof. Finally, how about a nod to Klein and Langmuir in the references?

We agree and have revised accordingly. This suggestion helped us considerably in sharpening and clarifying our arguments and conceptual model (Fig. 4 and relevant paragraphs). In particular, we highlight that the relevance of the heterogeneity being mantle derived is that it allows us to identify separate magma batches (whether both were stored

for a long period of time prior to eruption, or one is a new influx from the mantle), which then mix in a deep magma storage region. Thank you for this comment!

See lines 193 to 203.

Line 118-121: Isotopic “evolution” of the magmas: It is problematic that there are only 2 samples with all three radiogenic isotopic systems measured in them, and only 4 with both Sr and Nd. ALSO... the uncertainties reported in the supplement and shown on the images are unrealistically low, representing internal analytical standard errors, not true uncertainties. For instance, the reproducibility of 51 NBS 981 analyses reported as standard error, are much higher, and higher still, about 90 ppm, if unwrapped back to a standard deviation. The extremes of the Sr values are arguable indistinguishable given the standards data. Now, the apparent correlation with Nd isotopes in 4 samples is good, but this is by no means definitive. Especially in these sorts of generalist papers, it is important to emphasize what the state of the art, analytically speaking, is, and to honestly assess sources of uncertainty and reproducibility. I recall having to make similar arguments about barely resolvable Sr-Nd-Pb values at Kilauea about 20 years ago.

The reviewer makes three important and constructive points to address in this comment. We have made all the suggested changes. We fully agree that proper calculation and reporting of uncertainty is essential, and we have improved our efforts in this regard compared to the original submission. We hope the reviewer can now be confident that we are correctly resolving the trace element and isotope signature.

1) It is problematic that only two samples overlap between the Pb isotope and Sr and Nd isotope analyses.

We have addressed this by analyzing additional samples for Sr and Nd isotope ratios (now 7 sample analyses). The overlap is still not perfect (only 4 samples overlap), but our discussion does not rely on analysis of paired Sr-Nd-Pb isotope data. Moreover, within individual volcanic systems in Iceland strong correlations are seen between trace element ratios and radiogenic isotope ratios- the lower number of radiogenic isotope analyses allow us to confirm the involvement of both melting process and source heterogeneity in the generation of the two endmember melts.

2) All uncertainties are unrealistically low because only the internal instrumental standard error is used to estimate measurement uncertainty.

Throughout the manuscript we have recalculated all uncertainties for radiogenic isotope analyses to reflect reproducibilities of external standard reference materials, except where the internal standard error of the measurement was larger than the reproducibility of the reference materials. We always specify if we show 1 or 2 sigma external error.

Additionally, throughout the manuscript we have applied the same rigour to all other analyses, and now use external reproducibility of compositionally similar samples to define the uncertainty. Errors related to accuracy and precision are propagated to generate the

final calculated uncertainty. We now explicitly report our secondary standard data for all analyses (see Supplementary Tables).

3) The reproducibility of 51 NBS 981 analyses reported as standard error, are much higher, and higher still, about 90 ppm

We thank the reviewer for bringing our attention to this point. We now present the variation of reproduced standard data as standard deviation and have added additional standard information for the Sr and Nd isotope data.

Line 122-123: “been observed at other oceanic hotspots where”. I thought the authors were writing this as if this was a MOR eruption site, for which the Hawaiian comparison is not good. In fact, even in Hawaii, the comparison to Puu Oo is a particularly poor one. As a minimum the authors should explain the setting, the comparisons, and the salient points they are attempting to convey. It is quite a different thing to talk about mixing of source signatures at hot spots and at MORs, with perhaps the most applicable examples coming from plume-influenced ridges, such as the Galapagos Spreading Center and the HUMP region of the EPR, where individual eruption deposits have been mapped and samples for within flow variations (e.g., see Colman, A. et al. (2016) Magmatic Processes at Variable Magma Supply Along the Galápagos Spreading Center: Constraints from Individual Eruptive Units, *Journal of Petrology*, 57(5), 981-1018, doi: 10.1093/petrology/egw032 and Bergmanis, E., et al. (2007) Recent Eruptive History and Magma Reservoir Dynamics on the Southern East Pacific Rise at 17.5°S, *Geochem. Geophys. Geosyst.*, 8, Q12006, doi:10.1029/2007GC001742

We thank the reviewer for guiding us in this direction and, indeed, towards these relevant references (see also his suggestion below). To us, comparison of the observed (temporal) K_2O/TiO_2 variability at Fagradalsfjall is critical for highlighting the uniqueness of this eruption. The comparison in the manuscript has now been expanded extensively in a new supplementary figure, with the suggested literature dataset in mind. In the new supplementary figure (Extended Data Figure 12), we now bring in comparisons of K_2O/TiO_2 —that we adopt as a proxy for global geochemical enrichment (e.g., Jackson and Dasgupta, 2008)—comparing variability at Fagradalsfjall with well-characterized single-eruptive oceanic basalt units from (i) intraplate plumes (Puu Oo, Kilauea-2018), (ii) plume-influenced ridges (Galapagos spreading center), (iii) mid-ocean ridges (southern EPR) and (iv) recent MOR eruptions at the EPR (2005-2006) and Axial seamount. What is clear in all cases (including Puu Oo) is that the Fagradalsfjall eruption exhibits tremendous mantle-derived geochemical variability, highlighting its unique geochemical variability over a relatively short eruption.

The comparison to Puu Oo is made as it is one of the best monitored terrestrial basaltic eruptions, preserves mantle-derived geochemical variability, and so provides a critical point of comparison for the rate-of change of composition with the Fagradalsfjall eruption. We have been careful in the text to not imply a geodynamic comparison with Puu Oo.

Line 125-126: “However, the rapid compositional change in the case of the Fagradalsfjall eruption is unprecedented globally.”: I fundamentally disagree I don't think the authors are

not all that familiar with the literature.

The reviewer is correct that the change in major element chemistry is unremarkable globally; we emphasise that we are considering mantle-derived variability recorded by trace element ratios and radiogenic isotope ratios here (i.e., shifts in source compositions) and have modified the sentence to clarify this.

Line 130-135: I find this a very poor example for comparison, two primary reasons. a) the early phases of Puu oo have demonstrably been affected by crustal contamination with 1982 summit magma and b) it is unrealistic to compare a 35 year long, huge volume caldera driven rift zone eruption, which developed a buffered, steady state magma supply condition for all of the 90s, 2000s and 2010s, limiting the amount of variation.. Far better examples would be shorter events, such as the aforementioned 2018 LERZ eruption, or the aforementioned SEPR and GSC MOR eruptions, or other petrologically well studied individual MOR eruptions such as 9 50N EPR (Goss et al., G-cubed, 2010) or the last two at Axial Volcano (Clague, D. et al., (2018) Chemical Variations in the 1998, 2011, and 2015 Lava Flows from Axial Seamount, Juan de Fuca Ridge: Cooling During Ascent, Lateral Transport, and Flow, *Geochemistry, Geophysics, Geosystems*, 19, 2915–2933. <https://doi.org/10.1029/2018GC007708>.)

We thank the reviewer for guiding us in this direction. Please see our response to the previous comments

See lines 205 and 219 and Extended Data Figure 12.

Line 155-158: OHMI - Are we sure these unusual MI aren't in xenocrysts? Or, where might these melts sit and be picked up?

We agree and this is now pointed out explicitly in lines 177 to 180, where we write the following:

The presence of some macrocrysts too primitive to have crystallized from the Fagradalsfjall carrier liquid directly is consistent with an accumulated cognate load of crystals derived from mushes at the margins of a melt lens.

What is perhaps more important here is that we have added discussion regarding this point and how it fits with our observations overall and their implications for a magma genesis model (see also Figure 4c). We now discuss the possibility that some crystals carried by the Fagradalsfjall lava come from crystal/melt mushes at the margins of a melt lens, which we consider the most plausible explanation. Moreover, this observation, identified by the reviewer, has helped us construct what we regard to be a consistent model of magmagenesis. While some of these crystals and their inclusions may not be derived directly from the erupted melt, their geochemical correspondence with the host lavas demonstrates they are trapping similar melts that may have been present in the system in the years preceding eruption.

See lines 184 to 203.

2 Paragraphs, lines 160-188: This longish section could be greatly reduced to a sentence or two (as discussed in the last sentence of the 2nd paragraph), given the looseness of the constraints. In other words, it doesn't add that much to the paper, and it consumes a lot of space that could instead be used to describe some of the other attributes more fully, especially exoneration of extant literature on within eruption deposit mantle and crustal geochemical signals in MORB.

Agreed. We have followed the reviewer suggestion and shortened this text significantly.

See our response above and a completely revised section detailing the barometric model, lines 159-173

Line 206-207: " In the case of the latter, and in agreement with observations from global-scale variations in MORB" This could easily be elaborated upon, given the richness of the exiting MORB literature on the topic, and arguably, the limited applicability of observations at Holuhraun given the logical construct the authors themselves use, which is that Fagradalsfjall is a "MOR" like eruption where as Holuhraun was not MOR like. In fact, the arguments in ref 32 are based on a combination of global and individual eruption studies, one of the latter of which is described in detail, and which shows much greater within eruption compositional variability than the current example.

Agreed. We have followed the reviewer suggestion regarding this point and revised this part entirely.

See our response above and a revised discussion, lines 205 to 209.

Line 215-222: It would be better if the ranges and rates of variation in time were defined and described, for both this eruption, and other notable Icelandic and MOR examples, as studied for instance, by MacLennan and others. For instance, how do ranges of MI compositions vary with MgO as compared to Borgarhraun or the high Mg mamgas at Peistareykir?

Agreed. This would be very interesting – for a follow-up paper. In the interest of following the reviewer's primary suggestion that the manuscript be more focused, we have not pursued this topic. This is, of course, largely because this ancient eruption did not contain this exceptional timescale control on variation, which is indeed, almost impossible for eruptions at spreading ridges. Though we note that we plot melt inclusion K_2O/TiO_2 and La/Yb vs MgO in Extended Data Fig. 2 and give the range of K_2O/TiO_2 and La/Yb in the melt inclusions on edge of Figures 3a and 3b.

Line 224: "the entire spectrum..." This is the anticipated result in my opinion (as discussed for instance in ref 32), but nice to see it playing out in the data.

Agreed – it is very nice to see this so clearly in our data.

Line 237-239: I would be surprised if it weren't . But anyway, such a statement should be elaborated upon. How common or uncommon is it for collocated repeat eruptions from a restricted area to show completely different mantle source signatures?

At this juncture, we point out our new supplementary figure, which highlights the tremendous variability observed at the new Icelandic eruption site relative to other locations globally.

See responses above.

Line 252-253: “Second, for the first time, rapid partial drainage of a sub-Moho magma reservoir has been monitored in near-real time. “ Note, the 2005-6 EPR eruption described in the aforementioned Goss et al. paper was also hypothesized to have had a component of magma storage in a submoho reservoir identified and described by Dunn et al., based on seismic tomography.

We thank the reviewer for pointing this out. However, our main point here is this being the first near-real time observations of a rapid partial drainage of a near-Moho magma reservoir. Although the 2005–2006 eruptions at the East Pacific Rise (9°50'N) were discovered by ocean bottom seismometers, and several observations indicate that these eruptions likely occurred between mid-2005 to January 2006, it is our understanding these were not monitored in near-time (at least not from start to finish). Furthermore, we are not aware of petrological barometry being reported in the Goss et al. paper.

Reviewer Reports on the First Revision:

Referee #1:

I have reviewed the revised version of the manuscript and also reviewed the authors replies to my original review. In general, I feel like the authors have done a good job of answering the majority of my comments from the first time around, and tightening and improving the manuscript overall. I like the greater and more focused emphasis on this as an observed eruption that bypasses shallow crustal storage zones (which sees far more common in MOR), and also reveals very rapid shifts in magma composition. I also think that this will be of interest to a wide community of readers (particularly when coupled with the general high level of exposure this eruption received when it was happening).

I also appreciate that the authors have modified their interpretations (or the data used to make these) to ensure they primarily use samples and indices which reflect melt compositions. The role of inherited crystals is not explored greatly here – hopefully that will come as an important part of the story. However, the emphasis on incompatible elements and their ratios means that it is easier to see through potential complications related to crystal accumulation to the behavior of liquids. Likewise, I also appreciate how the interpretation of melt inclusion data is also now less dependent on accurate correction for post entrapment crystallization.

I also thank the authors for the inclusion of a large array of metadata to support the various geochemical measurements they report. The availability of this, including code used for barometry, makes this paper much better and more valuable to the community overall.

The section on barometry is improved, although one suggestion I have (at least for cpx barometry) is to also propagate analytical uncertainties for some selected cpx and see how the errors that this produces in pressure estimates compares to the error estimate they use, which comes from the calibration. It might be OK here as these cpx are relatively deep and thus the uncertainty in Na and Al (which drives much of the uncertainty in pressure related to analytical errors) is lower (as Na and Al are higher), but it would be worth checking that.

Some minor comments are:

Line 75 – give the depth here of mid to shallow crust (as this can mean different things in different settings)?

Line 85 – This is probably not the place to do this, but it is curious what the thinking behind the observation that initial seismicity seemed focused on a different ITSC

Line 138 – could the range of historical variation on the RP be put on Figure 3 as well?

Line 195 – is it possible that this mixing event actually triggered magma ascent into the shallower crust and eventual eruption.

Line 209-211. This sentence could use a reference – presumably the same as referred to below this.

Line 217-220. The meaning of the sentence starting “Even sites...” seems incomplete – maybe it could be rewritten

Line 225. The jump to an OIB system here (Kilauea) is maybe not necessary or needs some greater context to justify the comparison. I wonder if the improved documentation of MOR in the preceding paragraph is enough and the comparison to Kilauea – a different type of volcano in a different type of setting – may be less relevant.

Line 251 – how does this conclusion about the magma volume compare to geophysics, deformation or other information sources? If they are they are consistent this would be good supporting evidence.

Line 266 – diffusion studies (including some done by authors of this manuscript) in mantle minerals can provide insight into timescales of mantle processes such as mixing, so I think this sentence needs some modification.

Line 277. I think these are good questions, and another that is not raised in any detail here is how this eruption compares to MOR where the crust is much thinner. That seems key to address as we consider the ramifications of this study for MOR magmatism overall.

Fig 3. Are the uncertainties on K_2O/TiO_2 La/Yb really less than the size of the symbol here?

Referee #2:

This revised manuscript describing compositional changes in the first weeks of the 2021 Fagradalsfjall eruption in Iceland is an excellent example of a highly original, well-crafted, attentive to criticism, focused, and effective revision that I believe will be of high value and interest to the journal's readership. I fully endorse it for publication. I feel that in this revision, the authors have taken into consideration all of my prior criticisms and those of the other mail reviewer. They have also documented quite extensively in their rebuttal their collective thinking on these points, and how they are or aren't accommodated in the subsequent manuscript.

This is not to say that there aren't continuing small details that people could quibble with, and I believe the sub-editors may have some comments still about jargon, but I don't see these as being significant enough to even mention here in hopes that they don't delay what I anticipate will be the ultimate publication of the manuscript. I think the level of manuscript detail in describing the data, and the interpretations/implications of the data, is sufficient for the format. And of course as I stated in my prior review, the topic and observations are highly appropriate for the journal.

Having said all that I do want to point out one thing that the authors may wish to consider, which crystallized for me on the second reading of this much more focused and direct manuscript. In particular I am interested in the comment on line 219-220 regarding the relative amount of K/Ti and other types of variation, such as in radiogenic isotope ratios, in this eruption versus some of the

other well studied eruptions that are now referenced. This is an important observation and I'm not questioning the data, however I do wonder if the authors have considered alternate ways to perhaps explain some of the variation.

The way the authors present this is to suggest that somehow there is higher Fidelity in preserving greater mantle K/Ti variance in this case relative to the other tracers, as compared to other cited eruptions. It could be true, but I'm having a little bit of difficulty rationalizing how that would work unless there was inherently much more variation in K/Ti ratio in this case. Have the authors considered the alternate possibility that the other eruptions represent some sort of typical balance between those various compositional parameters as represented in the mantle, and that somehow this Icelandic eruption expresses enhanced, or extra variation in K and/or Ti, perhaps even from post-mantle processes?

I know many people, including some very prominent mid-ocean ridge basalt petrologist who I collaborate with frequently, are under the impression that K/Ti ratio is a good tracer of the mantle (as the current authors argue in manuscript lines 120/121). But I always like to remind folks of the caveat that this is only true in cases where melting or crystallization melt fraction does not get very low, because both of these elements, especially Ti, are susceptible to fractionation effects at very low melt fraction. A cryptic process such as in situ crystallization and compaction leading to release of low melt fraction liquids into a small magma body or dike from the associated cumulates might play a role in affecting this ratio in erupted liquids.

In other words, could it be possible that some amount of this enhanced K/Ti variation comes not from the mantle but from interaction at very low melt fraction with Crystal cumulates? And if so, what are the implications? I am thinking specifically about the types of effects one can find on increasing K/Ti ratio through interaction with a titaniferous phase, such as titanomagnetite which we know is common in Iceland, or even Kaersutitic amphiboles, which we know exist in some of the historical Icelandic bimodal compositional suites such as at Torfajökull (where I have myself found beautiful blue amphiboles with very high Ti).

There are probably some chemical tracers one could look at, for instance other transition metals, that might help establish whether or not a low melt fraction in situ crystallization style liquid could have been added to these magmas in the lower crust as they traversed. These titaniferous phases become stable at MgO concentrations as high as three or four wt percent (depending on other melting condition parameters such as pressure, water content, and oxygen fugacity). At even lower melt fraction and more differentiated conditions within a cumulate, it is also possible to affect potassium concentration through the presence of alkali feldspars and micas.

I guess what I'm saying is that in the interest of being comprehensive and covering their bases, I think that the authors may wish to either acknowledge this possibility of low melt fraction cumulate liquids on K/Ti, or present a very brief discussion and some backing calculations to show specifically that this possibility doesn't apply here. I would probably mention it on or near line 120 (e.g., that this is not considered important, and to refer readers to the supplement, where a few sentences could be added to the section on melt and glass compositions to describe the circumstances I just laid out, and support their assertions with some very basic calculations.

best wishes,
Ken Rubin

Other minor editorial notes:

Both the OPAM calculations and Saturation Pressure calculations supplements do not work (I get messages about errors in the source files when I try to download).

line 105: I would emphasize "shallow" magmatic plumbing systems (wherein crystal formation commonly tends to shift magmas to lower MgO).

line 111: insert "low pressure" before "magmatic"?

line 124-125. is this a sentence fragment? I am having difficulty following.

line 156. insert "partially" before "homogenized"

line 167: define macrocryst

line 181-183: is there a reason to not use the term "antecryst" and to cite the occurrence of these in other magmatic systems, including in Iceland?

line 219: qualify heterogeneity as 'radiogenic isotope and incompatible trace element heterogeneity'

Author Rebuttals to First Revision:

Referee #1:

I have reviewed the revised version of the manuscript and also reviewed the authors replies to my original review. In general, I feel like the authors have done a good job of answering the majority of my comments from the first time around, and tightening and improving the manuscript overall. I like the greater and more focused emphasis on this as an observed eruption that bypasses shallow crustal storage zones (which sees far more common in MOR), and also reveals very rapid shifts in magma composition. I also think that this will be of interest to a wide community of readers (particularly when coupled with the general high level of exposure this eruption received when it was happening).

I also appreciate that the authors have modified their interpretations (or the data used to make these) to ensure they primarily use samples and indices which reflect melt compositions. The role of inherited crystals is not explored greatly here – hopefully that will come as an important part of the story. However, the emphasis on incompatible elements and their ratios means that it is easier to see through potential complications related to crystal accumulation to the behavior of liquids. Likewise, I also appreciate how the interpretation of melt inclusion data is also now less dependent on accurate correction for post entrapment crystallization.

I also thank the authors for the inclusion of a large array of metadata to support the various geochemical measurements they report. The availability of this, including code used for barometry, makes this paper much better and more valuable to the community overall.

We thank Adam Kent for positive and constructive assessment of our work.

The section on barometry is improved, although one suggestion I have (at least for cpx barometry) is to also propagate analytical uncertainties for some selected cpx and see how the errors that this produces in pressure estimates compares to the error estimate they use, which comes from the calibration. It might be OK here as these cpx are relatively deep and thus the uncertainty in Na and Al (which drives much of the uncertainty in pressure related to analytical; errors) is lower (as Na and Al are higher), but it would be worth checking that.

As before, we fully agree with Adam Kent on the importance of rigorous assessment of the uncertainties in the barometry.

Regarding error propagation in the cpx-melt barometer associated with analytical uncertainties, we decided to follow his suggestions and check how instrumental error might influence error propagation, specifically with Na contents in mind as they define J_d in cpx. We selected a few cpxs crystals with variable Na₂O contents and, in accordance with our average instrumental error of 7% (see Supplementary data table S5), we added and subtracted 7% of their Na₂O and calculated the pressures with the melt-matching method again (see Methods). The max difference obtained was 0.6 kbar, which is well within the barometer's reported uncertainty.

See table below with examples of these calculations.

Example data using Na contents of cpx

Sample Name	Analysis name	Na ₂ O	Jd*		P**	1SD	
G20210321-1	G20210321-1-g4-coredarker	0.15	0.01		0.8	0.3	
G20210321-2	G20210321-2-g2-core	0.25	0.02		4.5	0.3	
G20210416-3	G20210416-3cpx43-3sd	0.20	0.01		2.9	0.4	
G20210416-3	G20210416-3cpx43-4sd	0.20	0.01		2.8	0.4	
G20210416-3	G20210416-3cpx44-MI	0.20	0.01		2.6	0.4	
G20210321-3	G20210321-3-g27-core	0.22	0.02		3.2	0.3	
G20210321-1	G20210321AC1_px1_incl_rimb	0.15	0.01		0.2	0.2	

Changed Na₂O content according to the instrumental error

					P	1SD	delta P***
+7% Na	G20210321-1-g4-coredarker+	0.16	0.01		1.3	0.3	0.5
	G20210321-2-g2-core+	0.27	0.02		4.7	0.3	0.2
	G20210416-3cpx43-3sd+	0.22	0.02		3.3	0.3	0.4
	G20210416-3cpx43-4sd+	0.22	0.02		3.3	0.3	0.5
	G20210416-3cpx44-MI+	0.21	0.02		3.1	0.3	0.5
	G20210321-3-g27-core+	0.23	0.02		3.7	0.3	0.5
	G20210321AC1_px1_incl_rimb+	0.16	0.01		0.8	0.2	0.6
-7% Na							
	G20210321-1-g4-coredarker-	0.14	0.01		0.4	0.3	-0.4
	G20210321-2-g2-core-	0.24	0.02		4.3	0.2	-0.2
	G20210416-3cpx43-3sd-	0.19	0.01		2.5	0.3	-0.4
	G20210416-3cpx43-4sd-	0.19	0.01		2.4	0.3	-0.4
	G20210416-3cpx44-MI-	0.18	0.01		2.2	0.3	-0.4
	G20210321-3-g27-core-	0.20	0.01		2.8	0.3	-0.4
	G20210321AC1_px1_incl_rimb-	0.14	0.01		0.0	0.2	-0.2

*calculated Jd content of clinopyroxene

**pressure outputs (in kbar) using clinopyroxene barometry methods described in methods

***difference between pressure output values (in kbar) after adding and subtracting representative instrumental error for Na contents

Some minor comments are:

Line 75 – give the depth here of mid to shallow crust (as this can mean different things in different settings)?

Agreed. We have added

i.e., the topmost 10 km of the crust to this sentence.

Line 85 – This is probably not the place to do this, but it is curious what the thinking behind the observation that initial seismicity seemed focused on a different ITSC

This certainly was an interesting observation, which in fact has been addressed in a paper that appeared in Nat Geoscience recently. See Flóvenz, Ó.G., Wang, R., Hersir, G.P. *et al.* Cyclical geothermal unrest as a precursor to Iceland's 2021 Fagradalsfjall eruption. *Nat. Geosci.* **15**, 397–404 (2022). <https://doi.org/10.1038/s41561-022-00930-5>

We now cite this paper.

Line 138 – could the range of historical variation on the RP be put on Figure 3 as well?

As the focus of these plots is on the temporal trend observed, we are worried that adding the range of historical variation on the RP on each of these panels will make them less accessible for readers. However, we have added two new panels to Extended Data Figure 4 which show how Fagradalsfjall compares with the range of historical variation on the RP. Furthermore, we can point out that Fagradalsfjall is also compared to historical RP lavas units on the following figures; Figure 2a, Extended Data Figure 2a and Extended Data Figure 13

Line 195 – is it possible that this mixing event actually triggered magma ascent into the shallower crust and eventual eruption.

Yes, this is a good suggestion and in fact, this is a good place to speculate on the importance of this mixing event. We have added this suggestion to this line in the revised manuscript:

It is therefore, well possible that this mixing event may have triggered magma ascent into the shallower crust and eventual eruption.

Line 209-211. This sentence could use a reference – presumably the same as referred to below this.

Agreed and reference 8 added after this sentence.

Line 217-220. The meaning of the sentence starting “Even sites...” seems incomplete – maybe it could be rewritten

Agreed. This sentence was incomplete (see also comment from Ken Rubin below). It has been modified and now reads:

*Even sites where single lava flows (e.g., the N1 unit at 17.5°S on the EPR) show both considerable radiogenic isotope and incompatible trace element heterogeneity²⁶ reveal limited K₂O/TiO₂ variability in comparison with Fagradalsfjall (**Extended data Figure 13**).*

Line 225. The jump to an OIB system here (Kilauea) is maybe not necessary or needs some greater context to justify the comparison. I wonder if the improved documentation of MOR in the preceding paragraph is enough and the comparison to Kilauea – a different type of volcano in a different type of setting – may be less relevant.

We do not agree on this point. Both OIB and MORB eruptions are used to study mantle-

derived variability, with Iceland representing a plume-influenced ridge, and therefore a mixture of both. We are describing an astonishing range of mantle-derived variability with unique timescale information. Therefore, we find it important to compare this with other key settings where we study similar things and Kilauea eruptions represent some exceptionally well-monitored basaltic eruptions.

Line 251 – how does this conclusion about the magma volume compare to geophysics, deformation or other information sources? If they are they are consistent this would be good supporting evidence.

This is a useful point to consider but it is a bit too early to say. The new study of Flóvenz et al., which we cite above (<https://doi.org/10.1038/s41561-022-00930-5>), estimate that 2.4–9 km³ of magma was required to produce the volume of CO₂ inferred as the trigger for seismicity in the adjacent ITSC. This is a huge volume, and is considerably larger than the volume of the reservoir we estimate (0.02 km³). However, the estimate by Flóvenz et al. is model dependent, reflects total input to the system (rather than the size of the chamber supplying the eruption), and relies on assumptions about the carbon-content of the mantle underlying Reykjanes (from which the primary CO₂ content of the magmas is estimated). On this last point, the Flóvenz et al. estimate does not include the contribution of the extremely carbon-rich primordial component that has been inferred beneath Iceland (Miller et al., EPSL, 2019), and the amount of carbon in the recycled component is unknown and could be considerably larger than assumed (Matthews et al., GCA, 2021). Since even the depleted endmember of the Fagradalsfjall lavas likely had some contribution from recycled components, it is likely that the Flóvenz et al. overestimated the volume of magma.

However, in another recently accepted manuscript (Pedersen et al., GRL, preprint available at <https://www.essoar.org/doi/10.1002/essoar.10509177.1>), changes in the lava effusion rate were used to estimate an initial melt reservoir volume of 0.02 km³ which coincides with our estimate based on magma mixing. We have therefore modified the sentence and it now reads:

*This is equivalent to a ~6 m thick disc of radius 1 km, consistent with estimates considering changes in effusion rates during the eruption¹⁶ and conceptual models of sill-like melt storage in the Icelandic lower crust*⁷.

Line 266 – diffusion studies (including some done by authors of this manuscript) in mantle minerals can provide insight into timescales of mantle processes such as mixing, so I think this sentence needs some modification.

Agreed. This sentence was modified. We now emphasize that our observations are direct and model-independent. It now reads;

Critically, studying mantle magma mixing through minerals and their MIs provides only indirect and model-dependent timescale and volumetric information.

Line 277. I think these are good questions, and another that is not raised in any detail here is how this eruption compares to MOR where the crust is much thinner. That seems key to

address as we consider the ramifications of this study for MOR magmatism overall.

Agreed. This sentence was modified but because of lack of space, we only added a few additional words highlight exactly this point. It now reads:

How widespread such deep magmatic plumbing system reconfigurations are at MOR and other oceanic islands where the crust is thinner, remains to be explored.

Fig 3. Are the uncertainties on K₂O/TiO₂ La/Yb really less than the size of the symbol here?

Yes, in the case of both ICP-OES and ICP-MS measurements, they are. For careful assessment of analytical uncertainties of these methods, see Supplementary Data Table, 2 and 3. Note that EPMA error is larger than this but to avoid the figure becoming too complex, we use the figure caption to guide readers to Extended Data Figure 2 where we show average 2sigma error associated with these measurements.

Referee #2:

This revised manuscript describing compositional changes in the first weeks of the 2021 Fagradalsfjall eruption in Iceland is an excellent example of a highly original, well-crafted, attentive to criticism, focused, and effective revision that I believe will be of high value and interest to the journal's readership. I fully endorse it for publication. I feel that in this revision, the authors have taken into consideration all of my prior criticisms and those of the other mail reviewer. They have also documented quite extensively in their rebuttal their collective thinking on these points, and how they are or aren't accommodated in the subsequent manuscript.

This is not to say that there aren't continuing small details that people could quibble with,

and I believe the sub-editors may have some comments still about jargon, but I don't see these as being significant enough to even mention here in hopes that they don't delay what I anticipate will be the ultimate publication of the manuscript. I think the level of manuscript detail in describing the data, and the interpretations/implications of the data, is sufficient for the format. And of course as I stated in my prior review, the topic and observations are highly appropriate for the journal.

We thank Ken Rubin for these kind words and his positive and constructive assessment of this work.

Having said all that I do want to point out one thing that the authors may wish to consider, which crystallized for me on the second reading of this much more focused and direct manuscript. In particular I am interested in the comment on line 219-220 regarding the relative amount of K/Ti and other types of variation, such as in radiogenic isotope ratios, in this eruption versus some of the other well studied eruptions that are now referenced. This is an important observation and I'm not questioning the data, however I do wonder if the authors have considered alternate ways to perhaps explain some of the variation. The way the authors present this is to suggest that somehow there is higher Fidelity in preserving greater mantle K/Ti variance in this case relative to the other tracers, as compared to other cited eruptions. It could be true, but I'm having a little bit of difficulty rationalizing how that would work unless there was inherently much more variation in K/Ti ratio in this case. Have the authors considered the alternate possibility that the other eruptions represent some sort of typical balance between those various compositional parameters as represented in the mantle, and that somehow this Icelandic eruption expresses enhanced, or extra variation in K and/or Ti, perhaps even from post-mantle processes?

I know many people, including some very prominent mid-ocean ridge basalt petrologist who I collaborate with frequently, are under the impression that K/Ti ratio is a good tracer of the mantle (as the current authors argue in manuscript lines 120/121). But I always like to remind folks of the caveat that this is only true in cases where melting or crystallization melt fraction does not get very low, because both of these elements, especially Ti, are susceptible to fractionation effects at very low melt fraction. A cryptic process such as in situ crystallization and compaction leading to release of low melt fraction liquids into a small magma body or dike from the associated cumulates might play a role in affecting this ratio in erupted liquids.

In other words, could it be possible that some amount of this enhanced K/Ti variation comes not from the mantle but from interaction at very low melt fraction with Crystal cumulates? And if so, what are the implications? I am thinking specifically about the types of effects one can find on increasing K/Ti ratio through interaction with a titaniferous phase, such as titanomagnetite which we know is common in Iceland, or even Kaersutitic amphiboles, which we know exist in some of the historical Icelandic bimodal compositional suites such as at Torfajökull (where I have myself found beautiful blue amphiboles with very high Ti).

There are probably some chemical tracers one could look at, for instance other transition metals, that might help establish whether or not a low melt fraction in situ crystallization style liquid could have been added to these magmas in the lower crust as they traversed.

These titaniferous phases become stable at MgO concentrations as high as three or four wt percent (depending on other melting condition parameters such as pressure, water content, and oxygen fugacity). At even lower melt fraction and more differentiated conditions within a cumulate, it is also possible to affect potassium concentration through the presence of alkali feldspars and micas.

I guess what I'm saying is that in the interest of being comprehensive and covering their bases, I think that the authors may wish to either acknowledge this possibility of low melt fraction cumulate liquids on K/Ti, or present a very brief discussion and some backing calculations to show specifically that this possibility doesn't apply here. I would probably mention it on or near line 120 (e.g., that this is not considered important, and to refer readers to the supplement, where a few sentences could be added to the section on melt and glass compositions to describe the circumstances I just laid out, and support their assertions with some very basic calculations.

This surely is an interesting topic and certainly worth exploring a bit. First, we would like to point out some key observations regarding K₂O/TiO₂ systematics of the Fagradalsfjall lavas.

1. While interactions between the magma and magmatic phases such as titanomagnetite and amphibole, which may fractionate K₂O/TiO₂, such phases are unlikely to present in substantial proportions in the magma storage region (e.g., Grove et al., 1992). In fact, it is clear that the Fagradalsfjall magma is too primitive to fractionate Ti-rich phases such as titanomagnetite which are common in low-MgO Icelandic basalts. No Ti-rich phases have been observed in the erupted products thus far, including a variety of nodules found. In addition, this magma is too primitive, too hot and too water-poor to stabilize amphibole to occur in evolved and alkalic Icelandic magmas. In other words, the Fagradalsfjall magma (including primitive inclusions) had low H₂O, halogen and TiO₂ contents and was fairly reduced, which means that the conditions for the saturation of amphibole or Ti-rich phases were not present.

To demonstrate this, we have constructed plots using the recently published Icelandic Volcanic rocks Isotopic Database (IVID) of Harðardóttir et al., (Spatial distribution and geochemical characterization of Icelandic mantle end-members: Implications for plume geometry and melting processes, Chemical Geology. 2022, <https://doi.org/10.1016/j.chemgeo.2022.120930>). See plots below

These figures show all Iceland lavas (filtered following protocols outlined in this paper, blue), all Reykjanes lavas (filtered, red), and Fagradalsfjall lavas (yellow). What is clear is that Reykjanes lavas shows no evidence of Ti-oxide fractionation: TiO₂ increases gradually with decreasing MgO, as expected. Even if some process was operating cryptically to exaggerate the range of K₂O/TiO₂ we observe, to see such coherent trends across Iceland would require this process to operate identically for every batch of magma. Fagradalsfjall lavas fall completely within the field for Reykjanes. What is also critical here is the fact that ²⁰⁶Pb/²⁰⁴Pb correlates with K₂O/TiO₂ across Reykjanes and that this relationship mirrors the correlation between ²⁰⁶Pb/²⁰⁴Pb and K₂O/TiO₂ at Fagradalsfjall – see next point.

2. K₂O/TiO₂ strongly correlates ($R^2 > 0.97$) with other tracers that must be mantle-derived (i.e., La/Yb or ²⁰⁶Pb/²⁰⁴Pb) – see plot below.

206Pb/204Pb and La/Yb are not affected by fractionation of Ti-rich phases and/or amphiboles, yet both ratios show a great range and are strongly correlated with K₂O/TiO₂. Hence, the observed K₂O/TiO₂ variability in the Fagradalsfjall lavas is therefore unlikely to be controlled by fractionation of Ti-rich phases and/or amphibole. This suggests that even if there is a secondary (crustal) process affecting the K₂O/TiO₂ values, the signal is almost entirely mantle-derived.

For this reason, but also to present arguments in a lucid manner, we rely on K₂O/TiO₂ in our work: it is a convenient tracer that we have measured in every sample. However, we emphasize - as these plots underline - that our arguments don't solely rest on K₂O/TiO₂.

- To further test the robustness of K₂O/TiO₂ as a proxy for mantle-derived variability for single eruptive units across the RP and Iceland, we constructed La/Yb plots equivalent to Figure 2 where we compare Fagradalsfjall K₂O/TiO₂ with this ratio in (a) historical lavas from RP and (b) single eruptive units from different parts of the Iceland rift system – see plot below.

What is clear from comparing Figure 2 in the manuscript with the La/Yb figures is that the large amplitude mantle-derived signal is almost identical in both figures. Again, this confirms that the K₂O/TiO₂ signal of Fagradalsfjall is primarily mantle-derived.

Following these suggestions from Ken Rubin, we decided to add this sentence to the lines he identified

While interactions between the magma and some magmatic phases (e.g., titanomagnetite and amphibole) can fractionate K₂O/TiO₂, such phases are unlikely to be present in substantial proportions in the magma storage region.

Moreover, we added two new panels to Extended Data Figure 4, that show how (i) K_2O/TiO_2 correlates directly with other tracers that must be mantle-derived (i.e., La/Yb or $^{206}Pb/^{204}Pb$) and (ii) Fagradalsfjall compares with the range of historical variation on the RP (per comment from Adam Kent above). To guide readers towards these new panels, we have added the following sentence,

Clear correlations ($R^2 > 0.97$) between K_2O/TiO_2 , La/Yb and Pb isotopes confirm that the large range in Fagradalsfjall K_2O/TiO_2 reflects mantle-derived variability.

best wishes,
Ken Rubin

Other minor editorial notes:

Both the OPAM calculations and Saturation Pressure calculations supplements do not work (I get messages about errors in the source files when I try to download).

We cannot find any issues with the files we provided, so we think this must be an issue with the Nature online manuscript handling system. These scripts are now stored in a Zenodo repository (<https://doi.org/10.5281/zenodo.6631328>); however, this is currently embargoed until publication of the manuscript. We will be happy to facilitate access to the code before publication.

line 105: I would emphasize "shallow" magmatic plumbing systems (wherein crystal formation commonly tends to shift magmas to lower MgO).

Agreed and done.

line 111: insert "low pressure" before "magmatic"?

Agreed and done.

line 124-125. is this a sentence fragment? I am having difficulty following.

Agreed and modified, it now reads;

Notably, it has been suggested that some of the other high-MgO units best preserve signatures associated with the Icelandic mantle (e.g., Borgarhraun, North Iceland)¹⁹.

line 156. insert "partially" before "homogenized"

Agreed and done.

line 167: define macrocryst

This term is defined in the supplement, line 44. To make this clear to our readers, we now guide them to the supplement.

line 181-183: is there a reason to not use the term "antecryst" and to cite the occurrence of these in other magmatic systems, including in Iceland?

As we see this, the term antecryst has a petrogenetic connotation- macrocryst is purely descriptive.

See Neave et al. (2012, JPet) for a discussion of this issue on terminology. They say:

“Using genetic terms such as phenocryst or xenocryst to describe crystals in volcanic rocks has significant limitations (Davidson et al., 2007; Ruprecht et al., 2012; Thomson & Maclennan, 2013). Observations of isotopic disequilibrium between crystals and their carrier liquids led Davidson et al. (2007) to suggest that crystals out of equilibrium with their surroundings, but nevertheless sourced from the same magmatic system, could be referred to as antecrysts (after W. Hildreth at the ‘Longevity and Dynamics of Rhyolitic Magma Systems’ Penrose Conference, 2001). However, the limits of a magmatic system are difficult to define, especially in Iceland where mantle melting has been generating oceanic crust of similar composition at the same location for millions of years (Thomson & Maclennan, 2013). The non-genetic term macrocryst is thus used throughout to refer to crystals with a minimum long axis length of 150 μm . This definition is based on the minimum size of crystals in rapidly quenched, glassy portions of thin sections.”

See <https://doi.org/10.1093/petrology/egu058>

We agree with the viewpoint presented by Neave et al, and prefer the term macrocryst where we are describing our observations.

line 219: qualify heterogeneity as 'radiogenic isotope and incompatible trace element heterogeneity'

Agreed and done.